# Costs and Benefits of Fair Regression

## Abstract

Real-world applications of machine learning tools in high-stakes domains are often regulated to be fair, in the sense that the predicted target should satisfy some quantitative notion of parity with respect to a protected attribute. However, the exact tradeoff between fairness and accuracy with a real-valued target is not entirely clear. In this paper, we characterize the inherent tradeoff between statistical parity and accuracy in the regression setting by providing a lower bound on the error of any attribute-blind fair regressor. Our lower bound is sharp, algorithm-independent, and admits a simple interpretation: when the moments of the target differ between groups, any fair algorithm has to make an error on at least one of the groups. We further extend this result to give a lower bound on the joint error of any (approximately) fair algorithm, using the Wasserstein distance to measure the quality of the approximation. With our novel lower bound, we also show that the price paid by a fair regressor that does not take the protected attribute as input is less than that of a fair regressor with explicit access to the protected attribute. On the upside, we establish the first connection between individual fairness, accuracy parity, and the Wasserstein distance by showing that if a regressor is individually fair, it also approximately verifies the accuracy parity, where the gap is again given by the Wasserstein distance between the two groups. Inspired by our theoretical results, we develop a practical algorithm for fair regression through the lens of representation learning, and conduct experiments on a real-world dataset to corroborate our findings.

## 1 Introduction

High-stakes domains, e.g., loan approvals, and credit scoring, have been using machine learning tools to help make decisions. A central question in these applications is whether the algorithm makes fair decisions, in the sense that certain sensitive data does not influence the outcomes or accuracy of the learning algorithms. For example, as regulated by the General Data Protection Regulation (GDPR, Article 22 Paragraph 4) (gdp), "decisions which produces legal effects concerning him or her or of similar importance shall not be based on certain personal data", including race, religious belief, etc. As a result, using the sensitive data directly in algorithm is often prohibited. However, due to the redundant encoding, redlining, and other problems, this "fairness through blindness" is often not sufficient to ensure algorithmic fairness in automated decision-making processes.

Many works have produced methods aiming at reducing unfairness (Calmon et al., 2017; Chi et al., 2021; Hardt et al., 2016; Agarwal et al., 2019; Feldman et al., 2015; Beutel et al., 2017; Lum & Johndrow, 2016) under various contexts. However, the question of the price that we need to pay for enforcing various fairness definitions in terms of the accuracy of these tools is less explored. In this paper, we attempt to answer this question by characterizing a tradeoff between statistical parity and accuracy in the regression setting, where the regressor is prohibited to use the sensitive attribute directly, dubbed as attribute-blind regressor (predictor). Among many definitions of fairness (Verma & Rubin, 2018) in the literature, statistical parity asks the predictor to be statistically independent of a predefined protected attribute, e.g., race, gender, etc. While empirically it has long been observed that there is an underlying tension between accuracy and statistical parity (Calders et al., 2013; Zliobaite, 2015; Berk et al., 2017; Agarwal et al., 2019) in both classification and regression settings, theoretical understanding of this tradeoff in regression is limited. In the case of classification, Menon & Williamson (2018) explored such tradeoff in terms of the fairness frontier function under the context of cost-sensitive binary classification. Zhao & Gordon (2019) provided a characterization of such tradeoff in binary classification. Recently, Chzhen et al. (2020a) and Le Gouic et al. (2020) concurrently with each other derived an analytic bound to characterize the price of statistical parity in regression using Wasserstein barycentres when the learner can take the sensitive attribute explicitly as an input.

In this paper, we derive the first lower bound to characterize the inherent tradeoff between fairness and accuracy in the regression setting under general $\ell_p$ loss when the regressor is prohibited to use the sensitive attribute directly during the inference stage. Our main theorem can be informally summarized as follows:

> *For* any *fair algorithm satisfying statistical parity, it has to incur a large error on at least one of the demographic subgroups when the moments of the target variable differ across groups. Furthermore, if the population of the two demographic subgroups are imbalanced, the minorities could still suffer from the reduction in accuracy even if the global accuracy does not seem to reduce.*

We emphasize that the above result holds in the noiseless setting as well, where there exist (unfair) algorithms that are perfect on both demographic subgroups. Hence it highlights the inherent tradeoff due to the coupling between statistical parity and accuracy in general, not due to the noninformativeness of the input. We also extend this result to the general noisy setting when only approximate fairness is required. Our bounds are algorithm-independent, and do not make any distributional assumptions. To illustrate the tightness of the lower bound, we also construct a problem instance where the lower bound is attained. In particular, it is easy to see that in an extreme case where the group membership coincides with the target task, a call for exact statistical parity will inevitably remove the perfect predictor. At the core of our proof technique is the use of the Wasserstein metric and its contraction property under certain Lipschitz assumption on the regression predictors.

On the positive side, we establish the first connection between individual fairness (Dwork et al., 2012), a more fine-grained notion of fairness, and accuracy parity (Buolamwini & Gebru, 2018; Bagdasaryan et al., 2019; Chi et al., 2021). Roughly speaking, an algorithm is said to be individually fair if it treats similar individuals similarly. We show that if a regressor is individually fair, then it also approximately verifies the accuracy parity. Interestingly, the gap in this approximation is exactly given by the Wasserstein distance between the distributions across groups. Our proof techniques are very simple but general, and we expect it to have broader applications in other learning scenarios with real target, e.g., domain adaptation for regression problems (Ganin et al., 2016; Courty et al., 2017; Zhao et al., 2019b) and counter-factual inference (Johansson et al., 2016; Shalit et al., 2017; Johansson et al., 2020).

Although our main focus is to understand the costs and benefits of fair regression, our analysis also naturally suggests a practical algorithm to achieve statistical parity and accuracy parity simultaneously in regression by learning fair representations. The idea is relatively simple and intuitive: it suffices if we can ensure that the representations upon which the regressor applies are approximately fair (measured by Wasserstein distance). Finally, we also conduct experiments on a real-world dataset to corroborate our theoretical findings. Our results highlight the role of the Wasserstein distance in both the theoretical analysis and algorithm design of fair regression, which complements the existing results for fair classification using TV-distance (Zhao & Gordon, 2022).

## 2 Preliminaries

**Notation** We consider a general regression setting where there is a joint distribution $\mu$ over the triplet $T = (X, A, Y)$, where $X \in \mathcal{X} \subseteq \mathbb{R}^d$ is the input vector, $A \in \{0, 1\}$ is the protected attribute, e.g., race, gender, etc., and $Y \in \mathcal{Y} \subseteq [-1, 1]$ is the target output.[1] Hence, the joint distribution $\mu$ is defined over the product space $\mathcal{X} \times \{0, 1\} \times \mathcal{Y}$. Lower case letters $\mathbf{x}$, $a$ and $y$ are used to denote the instantiation of $X$, $A$ and $Y$, respectively. Let $\mathcal{H}$ be a hypothesis class of predictors from input to output space. Throughout the paper, we focus on the setting where the regressor *cannot* directly use the sensitive attribute $A$ to form its prediction. However, note that even if the regressor does not explicitly take the protected attribute $A$ as input, this *fairness through blindness* mechanism can still be biased due to the redundant encoding issue (Barocas et al., 2017). To keep the notation uncluttered, for $a \in \{0, 1\}$, we use $\mu_a$ to mean the conditional distribution of $\mu$ given $A = a$. The zero-one entropy of $A$ (Grünwald et al., 2004, Section 3.5.3) is denoted as $H_{0\text{-}1}(A) := 1 - \max_{a \in \{0,1\}} \Pr(A = a)$. Furthermore, we use $F_\mu$ to represent the cumulative distribution function of $\mu$, i.e., for $t \in \mathbb{R}$, $F_\mu(t) := \Pr_\mu((-\infty, t])$. In this paper, we assume that the density of $\mu_i$ and its corresponding pushforward under proper transformation (w.r.t. the Lebesgue measure $\lambda$) is universally bounded above, i.e., $\|d\mu_i/d\lambda\|_\infty \leq C, \forall i \in \{0, 1\}$. Given a feature transformation function $g : \mathcal{X} \to \mathcal{Z}$ that maps instances from the input space $\mathcal{X}$ to feature space $\mathcal{Z}$, we define $g_\sharp \mu := \mu \circ g^{-1}$ to be the induced distribution (pushforward) of $\mu$ under $g$, i.e., for any measurable event $E' \subseteq \mathcal{Z}$, $\Pr_{g_\sharp \mu}(E') := \Pr_\mu(g^{-1}(E')) = \Pr_\mu(\{x \in \mathcal{X} \mid g(x) \in E'\})$. We also

---

[1]Our main results could be extended to the case where $A$ can take finitely many values.

use $Y_\sharp \mu$ to denote the marginal distribution of $Y$ from the joint distribution $\mu$, i.e., projection of $\mu$ onto the $Y$ coordinate. Throughout the paper, we make the following assumption:

**Assumption 2.1.** There exists a constant $C$ such that the density of $\mu$ (w.r.t. the Lebesgue measure $\lambda$) is universally bounded above, i.e., $\|d\mu/d\lambda\|_\infty \le C$.

**Fairness Definition** We mainly focus on group fairness where the group membership is given by the protected attribute $A$. In particular, *statistical parity* asks that the predictor should be statistically independent of the protected attribute. In binary classification, this requirement corresponds to the notion of equality of outcome (Holzer & Neumark, 2006), and it says that the outcome rate should be equal across groups.

**Definition 2.1** (Statistical Parity). Given a joint distribution $\mu$, a classifier $\widehat{Y} = h(X)$, satisfies *statistical parity* if $\widehat{Y}$ is independent of $A$.

Since $\widehat{Y}$ is continuous, the above definition implies that $\Pr_{\mu_0}(\widehat{Y} \in E) = \Pr_{\mu_1}(\widehat{Y} \in E)$ for any measurable event $E \subseteq \mathbb{R}$. Statistical parity has been adopted as definition of fairness in a series of work (Calders et al., 2009; Edwards & Storkey, 2015; Johndrow et al., 2019; Kamiran & Calders, 2009; Kamishima et al., 2011; Louizos et al., 2015; Zemel et al., 2013; Madras et al., 2018).

**Fair Regression** Given a joint distribution $\mu$, the weighted $\ell_p$ error of a predictor $\widehat{Y} = h(X)$ under $\mu$ for $p \ge 1$ is defined as

$$\varepsilon_{p,\mu}(\widehat{Y}) := \sum_a \Pr_\mu(A = a) \cdot \varepsilon_{p,\mu_a}(\widehat{Y}) = \sum_a \Pr_\mu(A = a) \cdot \left( \mathbb{E}_{\mu_a} \left[ |\widehat{Y} - Y|^p \right] \right)^{1/p}. \tag{1}$$

As two notable special cases, when $p = 2$, the above definition reduces to the square root of the usual mean-squared-error (MSE); when $p = 1$, (1) becomes the mean-absolute-error (MAE) of the predictor. To make the notation more compact, we may drop the subscript $\mu$ when it is clear from the context. The main departure from prior works on classification is that both $Y$ and $\widehat{Y}(h(X))$ are allowed to be real-valued rather than just categorical. Under statistical parity, the problem of fair regression (Agarwal et al., 2019) can be understood as the following constrained optimization problem:

$$
\begin{aligned}
\underset{h \in \mathcal{H}}{\text{minimize}} \quad & \varepsilon_{p,\mu}(\widehat{Y}) \\
\text{subject to} \quad & \left| \Pr_{\mu_0}(h(X) \le t) - \Pr_{\mu_1}(h(X) \le t) \right| \le \epsilon, \ \forall t \in \mathbb{R}.
\end{aligned}
\tag{2}
$$

Note that since $\widehat{Y} = h(X) \in \mathbb{R}$ is a real-valued random variable and $A$ is binary, the constraint in the above optimization formulation asks that the conditional cumulative distributions of $\widehat{Y}$ are approximately equal across groups, which is an additive approximation to the original definition of statistical parity. Formally, the constraint in (2) is known as the Kolmogorov-Smirnov distance:

**Definition 2.2** (Kolmogorov-Smirnov distance). For two probability distributions $\mu$ and $\mu'$ over $\mathbb{R}$, the *Kolmogorov-Smirnov distance* $K(\mu, \mu')$ is $K(\mu, \mu') := \sup_{z \in \mathbb{R}} |F_\mu(z) - F_{\mu'}(z)|$.

With the Kolmogorov-Smirnov distance, we can define the $\epsilon$-statistical parity for a regressor $h$:

**Definition 2.3** ($\epsilon$-Statistical Parity). Given a joint distribution $\mu$ and $0 \le \epsilon \le 1$, a regressor $\widehat{Y} = h(X)$, satisfies $\epsilon$-*statistical parity* if $K(h_\sharp \mu_0, h_\sharp \mu_1) \le \epsilon$.

Clearly, the slack variable $\epsilon$ controls the quality of approximation and when $\epsilon = 0$ it reduces to asking exact statistical parity as defined in Definition 2.1.

**Wasserstein Distance** Given two random variables $T$ and $T'$ with the corresponding distributions $\mu$ and $\mu'$, let $\Gamma(\mu, \mu')$ denote the set of all couplings $\gamma$ of $\mu$ and $\mu'$, i.e., $\gamma_T = \mu$ and $\gamma_{T'} = \mu'$. The *Wasserstein distance* between the pair of distributions $\mu$ and $\mu'$ is defined as follows:

$$W_p(\mu, \mu') := \left( \inf_{\gamma \in \Gamma(\mu, \mu')} \int \|T - T'\|^p \, d\gamma \right)^{1/p}, \tag{3}$$

where $p \geq 1$ and throughout this paper we fix $\|\cdot\|$ to be the $\ell_2$ norm. For the special case where both $\mu$ and $\mu'$ are distributions over $\mathbb{R}$, the Wasserstein distance $W_p(\mu, \mu')$ admits the following equivalent characterization (Kolouri et al., 2017):

$$W_p(\mu, \mu') = \left( \int_0^1 |F_\mu^{-1}(t) - F_{\mu'}^{-1}(t)|^p \, dt \right)^{1/p}, \tag{4}$$

where $F_\mu^{-1}(t)$ denotes the generalized inverse of the cumulative distribution function, i.e., $F_\mu^{-1}(t) = \inf_{z \in \mathbb{R}}\{z : F(z) \geq t\}$. The above closed form formulation will be particularly useful in our later analysis. When $p = 1$, the Wasserstein distance is also called the *Earth Mover distance*, and it admits a dual representation in a variational form using sup rather than inf: $W_1(\mu, \mu') = \sup_{f : \|f\|_L \leq 1} |\int f \, d\mu - \int f \, d\mu'|$, where $\|f\|_L := \sup_{\mathbf{x} \neq \mathbf{x}'} |f(\mathbf{x}) - f(\mathbf{x}')| / |\mathbf{x} - \mathbf{x}'|$ is the Lipschitz seminorm of $f$. It is well-known that convergences of measures under the Wasserstein distance implies weak convergence, i.e., convergence in distribution (Gibbs & Su, 2002). Furthermore, compared with other distance metrics including total variation (TV), Jensen-Shannon distance, etc. that ignore the geometric structure of the underlying space, Wasserstein distance often allows for more robust applications, e.g., the Wasserstein GAN (Arjovsky et al., 2017), domain adaptation (Courty et al., 2017), etc., due to the its Lipschitz continuous constraint in the dual representation. Moreover, unlike the KL divergence, the Wasserstein distance between two measures is generally finite even when neither measure is absolutely continuous with respect to the other, a situation that often arises when considering empirical distributions arising in practice. Furthermore, unlike the TV-distance, the Wasserstein distance inherently depends on the geometry of the underlying space, whereas the TV distance is invariant under any bijective mapping.

## 3  Main Results

Recently, Agarwal et al. (2019) proposed a reduction-based approach to tackle (2) by solving a sequence of cost-sensitive problems. By varying the slack variable $\epsilon$, the authors also empirically verified the unavoidable tradeoff between statistical parity and accuracy in practice. However, to the best of our knowledge, a quantitative characterization on the exact tradeoff between fairness and accuracy is still missing. In this section, we seek to answer the following intriguing and important question:

> *In the setting of regression, what is the minimum error that any attribute-blind fair algorithm has to incur, and how does this error depend on the coupling between the target and the protected attribute?*

In what follows we shall first provide a simple example to illustrate this tradeoff. This example will give readers a flavor the kind of impossibility result we are interested in proving. We then proceed to formally present our first theorem which exactly answers the above question, even if only approximate fairness is satisfied. We conclude this section by some discussions on the implications of our results.

**A Simple Example**  As a warm-up, let us consider an example to showcase the potential tradeoff between statistical parity and accuracy. But before our construction, it should be noted that the error $\varepsilon_{p,\mu}(\widehat{Y})$ bears an intrinsic lower bound for any deterministic predictor $\widehat{Y} = h(X)$, i.e., the noise in the underlying data distribution $\mu$. Hence to simplify our discussions, in this example we shall construct distributions such that there is no noise in the data, i.e., for $a \in \{0, 1\}$, there exists a ground-truth labeling function $h_a^*$ such that $Y = h_a^*(X)$ on $\mu_a$. Realize that such simplification will only make it harder for us to prove lower bound on $\varepsilon_{p,\mu_a}$ since there exists predictors that are perfect.

**Example 3.1** (Target coincides with the protected attribute). For $a \in \{0, 1\}$, let the marginal distribution $X_\sharp \mu_a$ be a uniform distribution over $\{0, 1\}$. Let $Y = a$ be a constant. Hence by construction, on the joint distribution, we have $Y = A$ hold. Now for any fair predictor $\widehat{Y} = h(X)$, the statistical parity asks $\widehat{Y}$ to be independent of $A$. However, no matter what value $h(x)$ takes, we always have $|h(x)| + |h(x) - 1| \geq 1$. Hence for any predictor $h : \mathcal{X} \to \mathbb{R}$:

$$\varepsilon_{1,\mu_0}(h) + \varepsilon_{1,\mu_1}(h) = \frac{1}{2}|h(0) - 0| + \frac{1}{2}|h(1) - 0| + \frac{1}{2}|h(0) - 1| + \frac{1}{2}|h(1) - 1| \geq \frac{1}{2} + \frac{1}{2} = 1.$$

This shows that for any fair predictor $h$, the sum of $\ell_1$ errors of $h$ on both groups has to be at least 1. On the other hand, there exists a trivial unfair algorithm that makes no error on both groups by also taking the protected attribute into consideration: $\forall x \in \{0, 1\}, h^*(x) = 0$ if $A = 0$ else $h^*(x) = 1$.

## 3.1 The Cost of Statistical Parity under Noiseless Setting

The example in the previous section corresponds to a worst case where $Y = A$. On the other hand, it is also clear that when the target variable $Y$ is indeed independent of the protected attribute $A$, there will be no tension between statistical parity and accuracy. The following theorem exactly characterizes the tradeoff between fairness and accuracy by taking advantage of the relationship between $Y$ and $A$:

**Theorem 3.1.** Let $\widehat{Y} = h(X)$ be a predictor. If $\widehat{Y}$ satisfies statistical parity, then $\forall p \geq 1$,

$$\varepsilon_{p,\mu_0}(\widehat{Y}) + \varepsilon_{p,\mu_1}(\widehat{Y}) \geq W_p(Y_\sharp\mu_0, Y_\sharp\mu_1). \tag{5}$$

We provide a proof by picture to illustrate the high-level idea of the proof in Fig. 1. For the special case of $p = 1$ and $p = 2$, Theorem 3.1 gives the following lower bounds on the sum of MAE and MSE on both groups respectively:

**Corollary 3.1.** If $\widehat{Y}$ satisfies statistical parity, then $\varepsilon_{1,\mu_0}(\widehat{Y}) + \varepsilon_{1,\mu_1}(\widehat{Y}) \geq |\mathbb{E}_{\mu_0}[Y] - \mathbb{E}_{\mu_1}[Y]|$ and $\varepsilon_{2,\mu_0}^2(\widehat{Y}) + \varepsilon_{2,\mu_1}^2(\widehat{Y}) \geq \frac{1}{2}|\mathbb{E}_{\mu_0}[Y] - \mathbb{E}_{\mu_1}[Y]|^2$.

**Remark** First of all, the lower bound $W_p(Y_\sharp\mu_0, Y_\sharp\mu_1)$ corresponds to a measure of the distance between the marginal distributions of $Y$ conditioned on $A = 0$ and $A = 1$ respectively. Hence when $A$ is independent of $Y$, we will have $Y_\sharp\mu_0 = Y_\sharp\mu_1$ so that the lower bound gracefully reduces to 0, i.e., no essential tradeoff between fairness and accuracy. On the other extreme, consider $Y = cA$, where $c > 0$. In this case $A$ fully describes $Y$ and it is easy to verify that $W_p(Y_\sharp\mu_0, Y_\sharp\mu_1) = c$, which means the lower bound also takes into account the magnitude of the target variable $Y$. For a protected attribute $A$ that takes more than 2 values, we could extend Theorem 3.1 by considering all possible pairwise lower bounds and average over them. Furthermore, the lower bound is sharp, in the sense that for every $p \geq 1$, there exist problem instances that achieve the above lower bound, e.g., Example 3.1. To see this, consider the $\ell_p$ error for any predictor $h$, we have

$$\begin{aligned}
\varepsilon_{p,\mu_0}(h) + \varepsilon_{p,\mu_1}(h) &= \left(\frac{1}{2}|h(0) - 0|^p + \frac{1}{2}|h(1) - 0|^p\right)^{1/p} + \left(\frac{1}{2}|h(0) - 1|^p + \frac{1}{2}|h(1) - 1|^p\right)^{1/p} \\
&\geq \frac{1}{2}|h(0) - 0| + \frac{1}{2}|h(1) - 0| + \frac{1}{2}|h(0) - 1| + \frac{1}{2}|h(1) - 1| \\
&\geq \frac{1}{2} + \frac{1}{2} = 1 = W_p(Y_\sharp\mu_0, Y_\sharp\mu_1),
\end{aligned}$$

where the first inequality follows from Jensen's inequality and the fact that the $\ell_p$ norm is convex. On the other hand, it can be readily verified that the fair predictor $h(X) = 1/2$ attains the lower bound. As another example, consider the following Gaussian case for $p = 2$:

**Example 3.2** (Gaussian case). For $a \in \{0, 1\}$, let the marginal distribution $X_\sharp\mu_a$ be a standard Gaussian distribution $\mathcal{N}(0, I_d)$ and assume $A \perp X$. Fix $w \in \mathbb{R}^d$ with $\|w\| = 1$, and construct $Y_0 = w^T X - 1$ and $Y_1 = w^T X + 1$. Now for any regressor $\widehat{Y} = h(X)$, due to the data-processing inequality, $\widehat{Y} \perp A$ so $\widehat{Y}$ is fair. However, consider the $\ell_2$ error of $h$ on both groups:

$$\begin{aligned}
\varepsilon_{2,\mu_0}(h) + \varepsilon_{2,\mu_1}(h) &= \mathbb{E}_X^{1/2}[(h(X) - Y_0)^2] + \mathbb{E}_X^{1/2}[(h(X) - Y_1)^2] \\
&\geq \mathbb{E}_X[|h(X) - Y_0|] + \mathbb{E}_X[|h(X) - Y_1|] \geq \mathbb{E}_X[|Y_0 - Y_1|] = 2,
\end{aligned}$$

where the first inequality is due to Jensen's inequality. On the other hand, note that the distributions of $Y_0$ and $Y_1$ are $\mathcal{N}(-1, 1)$ and $\mathcal{N}(1, 1)$, respectively. The analytic formula (Givens et al., 1984, Proposition 7) for the $W_2$ distance between two Gaussians $\mathcal{N}(m_0, \Sigma_0)$ and $\mathcal{N}(m_1, \Sigma_1)$ is

$$W_2^2(\mathcal{N}(m_0, \Sigma_0), \mathcal{N}(m_1, \Sigma_1)) = \|m_0 - m_1\|^2 + \text{Tr}\left(\Sigma_0 + \Sigma_1 - 2\left(\Sigma_0^{1/2}\Sigma_1\Sigma_0^{1/2}\right)^{1/2}\right),$$

which shows that $W_2(Y_0, Y_1) = |-1 - 1| = 2$. Further, consider $\widehat{Y}^* = h^*(X) = w^T X$, then

$$\varepsilon_{2,\mu_0}(h^*) + \varepsilon_{2,\mu_1}(h^*) = \mathbb{E}_X^{1/2}[(h^*(X) - Y_0)^2] + \mathbb{E}_X^{1/2}[(h^*(X) - Y_1)^2] = 1 + 1 = 2.$$

Hence $h^*$ achieves the lower bound and the lower bound is verified.

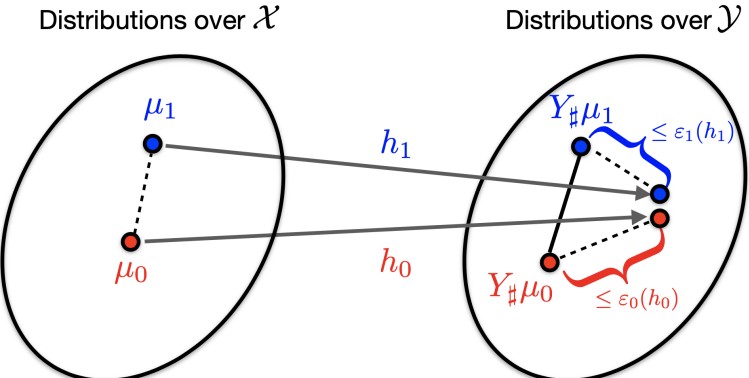

Figure 1: Proof by picture. The predictors on $\mu_0$ and $\mu_1$ induce two predictive distributions over $\mathcal{Y}$. Applying a chain of triangle inequalities (with $W_p(\cdot, \cdot)$) to the quadrilateral in the right circle and using the fact that the Wasserstein distance is a lower bound of the regression error then completes the proof. Note that here the predictors $h_0$ and $h_1$ over $\mu_0$ and $\mu_1$ need not to be the same.

It is worth pointing out that the lower bound in Theorem 3.1 is algorithm-independent and it holds on the population distribution. That being said, by using recent tail bounds (Lei et al., 2020; Weed et al., 2019) on the expected Wasserstein distance between the empirical distributions and its population counterpart, it is not hard to extend Theorem 3.1 to obtain a finite sample high probability bound of Theorem 3.1:

**Theorem 3.2.** Let $\widehat{Y} = h(X)$ be the predictor and $\hat{\mu}$ be an empirical distribution induced from a sample of size $n$ drawn from $\mu$. If $\widehat{Y}$ satisfies statistical parity, then there exists an absolute constant $c_1 > 0$ such that for $0 < \delta < 1$, with probability at least $1 - \delta$ over the draw of the sample,

$$\varepsilon_{2,\mu_0}(\widehat{Y}) + \varepsilon_{2,\mu_1}(\widehat{Y}) \geq \varepsilon_{1,\mu_0}(\widehat{Y}) + \varepsilon_{1,\mu_1}(\widehat{Y}) \geq W_1(Y_\sharp\hat{\mu}_0, Y_\sharp\hat{\mu}_1) - \left(2c_1 + \sqrt{2\log(2/\delta)}\right)\sqrt{\frac{1}{n}}. \tag{6}$$

**Remark** It is possible to obtain better lower bounds for the $\ell_2$ error in Theorem 3.2, but that requires making more assumptions on the underlying distribution $\mu$, e.g., strongly log-concave density. The first term in the lower bound, $W_1(Y_\sharp\hat{\mu}_0, Y_\sharp\hat{\mu}_1)$, could be efficiently computed from the data by solving a linear program (Cuturi & Doucet, 2014, Problem (3)). Furthermore, it is worth pointing out that the lower bound in Theorem 3.2 applies to all the predictors $\widehat{Y}$ and is insensitive to the marginal distribution of $A$. As a comparison, let $p_a := \Pr_\mu(A = a)$, then $\varepsilon_{p,\mu}(\widehat{Y}) = p_0\varepsilon_{p,\mu_0}(\widehat{Y}) + p_1\varepsilon_{p,\mu_1}(\widehat{Y})$. In this case if the group ratio is imbalanced, the overall error $\varepsilon_{p,\mu}(\widehat{Y})$ could still be small even if the minority group suffers a large error. Using Theorem 3.1, we can also bound the joint error over all the population:

**Corollary 3.2.** Let $\widehat{Y} = h(X)$ be a predictor. If $\widehat{Y}$ satisfies statistical parity, then $\forall p \geq 1$, the joint error has the following lower bound:

$$\varepsilon_{p,\mu}(\widehat{Y}) \geq H_{0\text{-}1}(A) \cdot W_p(Y_\sharp\mu_0, Y_\sharp\mu_1). \tag{7}$$

Compared with the one in Theorem 3.1, the lower bound of the joint error in Corollary 3.2 additionally depends on the zero-one entropy of $A$. In particular, if the marginal distribution of $A$ is skewed, then $H_{0\text{-}1}(A)$ will be small, which means that fairness will not reduce the joint accuracy too much. In this case, even if $W_p(Y_\sharp\mu_0, Y_\sharp\mu_1)$ is large, it might seem like that the joint error $\varepsilon_{p,\mu}(\widehat{Y})$ need not be large. However, this is due to the fact that the price in terms of the drop in accuracy is paid by the minority group. Our observation here suggests that the joint error $\varepsilon_{p,\mu}(\widehat{Y})$ is not necessarily the objective to look at in high-stakes applications, since it naturally encodes the imbalance between different subgroups into account. Instead, a more appealing alternative to consider is the *balanced error rate*:

$$\text{Balanced Error Rate of } \widehat{Y} := \frac{1}{2}\left(\varepsilon_{p,\mu_0}(\widehat{Y}) + \varepsilon_{p,\mu_1}(\widehat{Y})\right), \tag{8}$$

which applies balanced weights to both groups in the objective function. Clearly, (8) could be reduced to the so-called *cost-sensitive loss*, where data from group $a \in \{0, 1\}$ is multiplied by a positive weight that is reciprocal to the group's population level, i.e., $1/\Pr(A = a)$.

**Comparisons with Related Lower Bounds**   It is instructive to compare the above lower bound for the population error with the one of (Chzhen et al., 2020a, Theorem 2.3), where the authors use a Wasserstein barycenter characterization to give a lower bound on the special case of squared $\ell_2$ error ($p = 2$) when the regressor can explicitly take the protected attribute as its input. As a comparison, our results apply to the general $\ell_p$ loss. To provide a more formal and detailed comparison, we first state the theorem for the mean-squared error in the setting where the regressor has explicit access to the protected attribute $A$ from Chzhen et al. (2020a) (using adapted notation for consistency):

**Theorem 3.3** (Chzhen et al. (2020a) Theorem 2.3). Assume, for each $a \in \{0,1\}$, that the univariate measure $Y_\sharp \mu_a$ has a density and let $p_a := \Pr(A = a)$. Then,

$$\min_{g \text{ satisfies statistical parity}} \mathbb{E}[(f^*(X, A) - g(X, A))^2] = \min_\nu \sum_{a \in \{0,1\}} p_a \cdot W_2^2(Y_\sharp \mu_a, \nu),$$

where $f^*(X, A)$ is the Bayes optimal regressor and $\nu$ is a distribution over $\mathbb{R}$.

**Remark**   First, the quantity of interest in Theorem 3.3 is the discrepancy between a fair predictor $g(\cdot, \cdot)$ and the Bayes optimal predictor $f^*(\cdot, \cdot)$. On the other hand, the cost we are interested in this work is the excess risk (c.f. Definition 3.1) of a fair predictor. These two terms are not the same in general. However, in the noiseless setting, we know that $Y = f^*(X, A)$ and the excess risk reduces to the error $\varepsilon_{2,\mu}$. In this case, the costs of fairness in Theorem 3.1 and Theorem 3.3 has the following relationship:

$$\mathbb{E}[(Y - g(X, A))^2] = \sum_{a \in \{0,1\}} p_a \cdot \varepsilon_{2,\mu_a}^2(\widehat{Y}) = \left( \sum_{a \in \{0,1\}} p_a \cdot \varepsilon_{2,\mu_a}^2(\widehat{Y}) \right) \cdot \left( \sum_{a \in \{0,1\}} p_a \right)$$

$$\geq \left( \sum_{a \in \{0,1\}} \sqrt{p_a} \cdot \sqrt{p_a} \varepsilon_{2,\mu_a}(\widehat{Y}) \right)^2 = \varepsilon_{2,\mu}^2(\widehat{Y}).$$

Furthermore, although in both Theorem 3.1 and Theorem 3.3 we require the predictor to be fair in the sense of statistical parity, the class of feasible predictors in Theorem 3.3 is still larger than that of Theorem 3.1. To see this, note that in Theorem 3.1, beyond asking for $h(\cdot)$ to be fair, the same attribute-blind predictor $h$ has to be used in both groups. On the other hand, although the attribute-aware predictor $g(\cdot, \cdot)$ is constrained to be fair, different predictors $g(\cdot, a)$ could be applied over different groups indexed by $A = a$. Hence, even in the noiseless case with $p = 2$, the two lower bounds in Theorem 3.1 and Theorem 3.3 are not directly comparable, and one cannot be used to deduce the other.

Another related result in the literature is Theorem 3.1 of Chi et al. (2021), which we restate as follows:

**Theorem 3.4** (Chi et al. (2021) Theorem 3.1). Let $\widehat{Y} = h(X)$ be a predictor, then

$$\varepsilon_{2,\mu_0}^2(\widehat{Y}) + \varepsilon_{2,\mu_1}^2(\widehat{Y}) \geq \frac{1}{2} \left( \left[ W_1(Y_\sharp \mu_0, Y_\sharp \mu_1) - W_1(h_\sharp \mu_0, h_\sharp \mu_1) \right]_+ \right)^2,$$

where $[t]_+ := \max\{0, t\}$.

**Remark**   When $h(\cdot)$ satisfies the statistical parity exactly, the above lower bound reduces to $\frac{1}{2} W_1(Y_\sharp \mu_0, Y_\sharp \mu_1)^2$. Again, Theorem 3.4 is not directly comparable to Theorem 3.1 due to the same reason that Theorem 3.1 and Theorem 3.3 are not directly comparable. However, we can compare Theorem 3.4 and Theorem 3.3 directly since both costs are measured by the mean-squared error. To do so, realize that for the special case of $W_2$ with $\|\cdot\|_2$ as the underlying metric, we know that the Wasserstein barycenter lies on the Wasserstein geodesic between $Y_\sharp \mu_0$ and $Y_\sharp \mu_1$ (Villani, 2009). Let $\nu^* = \arg\min \sum_{a \in \{0,1\}} p_a \cdot W_2^2(Y_\sharp \mu_a, \nu)$, i.e., $\nu^*$ is the Wasserstein barycenter. Now since $W_2(\cdot, \cdot)$ is a metric and $\nu^*$ lies on the geodesic, we know

$$W_2(Y_\sharp \mu_0, \nu^*) + W_2(Y_\sharp \mu_1, \nu^*) = W_2(Y_\sharp \mu_0, Y_\sharp \mu_1).$$

By using this fact, it is straightforward to verify that the following chain of inequalities holds:

$$W_2^2(Y_\sharp \mu_0, \nu^*) + W_2^2(Y_\sharp \mu_1, \nu^*) \geq \frac{1}{2} \left( W_2(Y_\sharp \mu_0, \nu^*) + W_2(Y_\sharp \mu_1, \nu^*) \right)^2 = \frac{1}{2} W_2^2(Y_\sharp \mu_0, Y_\sharp \mu_1) \geq \frac{1}{2} W_1^2(Y_\sharp \mu_0, Y_\sharp \mu_1),$$

where the first inequality is due to the AM-GM inequality and the last one is due to the monotonicity of the $W_p(\cdot, \cdot)$ metric. Hence, in the special case of $p = 2$ with mean-squared error, the lower bound in Theorem 3.3 is tighter than the one in Theorem 3.4.

### 3.2 Extension to Approximate Fairness under Noisy Setting

In the previous section we show that there is an inherent tradeoff between statistical parity and accuracy when a predictor *exactly* satisfies statistical parity, and in particular this holds even if there is a perfect (unfair) regressor in both groups, i.e., there is no noise in the underlying population distribution. However, as formulated in (2), in practice we often only ask for approximate fairness where the quality of approximation is controlled by the slack variable $\epsilon$. Furthermore, even without the fairness constraint, in most interesting problems we often cannot hope to find perfect predictors for the regression problem of interest. Hence, it is natural to ask what is the tradeoff between fairness and accuracy when our predictor only approximately satisfies fairness ($\epsilon$-SP, Definition 2.3) over general distribution $\mu$?

In this section we shall answer this question by generalizing our previous results to prove lower bounds on both the sum of conditional and the joint target errors that also take the quality of such approximation into account. Due to potential noise in the underlying distribution, we first define the excess risk $r_{p,\mu}(\widehat{Y})$ of a predictor $\widehat{Y}$, which corresponds to the reducible error:

**Definition 3.1** (Excess Risk). Let $\widehat{Y} = h(X) \in \mathbb{R}$ be a predictor. The $\ell_p$ excess risk of $\widehat{Y}$ is defined as $r_{p,\mu}(\widehat{Y}) := \varepsilon_{p,\mu}(\widehat{Y}) - \varepsilon_{p,\mu}^*$, where $\varepsilon_{p,\mu}^* := \inf_f \varepsilon_{p,\mu}(f(X))$ is the optimal error over all measurable functions.

Assuming the infimum is achievable, we use $f_i^*$ to denote an optimal regressor without fairness constraint over $\mu_i, i \in \{0,1\}$, i.e., $f_i^* \in \arg\min_f \varepsilon_{p,\mu_i}(f(X))$. Then we have the following hold:

**Proposition 3.1.** Let $\widehat{Y} = h(X)$ be a predictor. Under Assumption 2.1, for $p \geq 1$, if there exists $\epsilon > 0$ such that $W_p(h_\sharp \mu_0, h_\sharp \mu_1) \leq \epsilon$, then

$$r_{p,\mu_0}(\widehat{Y}) + r_{p,\mu_1}(\widehat{Y}) \geq \underbrace{W_p(f_{0\sharp}^*\mu_0, f_{1\sharp}^*\mu_1)}_{\text{distance between optimal unfair predictors across groups}} -2(\varepsilon_{p,\mu_0}^* + \varepsilon_{p,\mu_1}^*) - \epsilon, \tag{9}$$

and $\widehat{Y}$ satisfies $2\sqrt{C\epsilon}$-SP.

**Remark**   It is easy to verify that Proposition 3.1 is a generalization of the lower bound presented in Theorem 3.1: when $f_i^*$ are perfect predictors, we have $Y_\sharp \mu_i = f_{i\sharp}^* \mu_i$ and $\varepsilon_{p,\mu_i}^* = 0$, for $i \in \{0,1\}$. Hence in this case the excess risk $r_{p,\mu_i}(\widehat{Y})$ reduces to the error $\varepsilon_{p,\mu_i}(\widehat{Y})$. Furthermore, if $\epsilon = 0$, i.e., $\widehat{Y}$ satisfies the exact statistical parity condition, then the lower bound (9) recovers the lower bound (5). As a separate note, Proposition 3.1 also implies that one can use the Wasserstein distance between the predicted distributions across groups as a proxy to ensure approximate statistical parity. This observation has also been shown in Dwork et al. (2012, Theorem 3.3) in classification.

### 3.3 Individual Fairness, Accuracy Parity and the Wasserstein Distance

In the previous section we show that the Wasserstein distance between the output distributions across groups could be used as a proxy to ensure approximate statistical parity. Nevertheless, Theorem 3.1 and Proposition 3.1 show that statistical parity is often at odds with the accuracy of the predictor, and in many real-world scenarios SP is insufficient to be used as a notion of fairness (Dwork et al., 2012, Section 3.1). Alternatively, in the literature a separate notion of fairness, known as *individual fairness*, has been proposed in Dwork et al. (2012). Roughly speaking, under the framework of individual fairness, for the classification task $T$ of interest at hand, the learner will have access to a (hypothetical) task-specific metric $d_T(\cdot, \cdot)$ for determining the degree to which individuals are similar w.r.t. the task $T$. We emphasize here that the metric $d_T(\cdot, \cdot)$ should be task-specific, and in practice it is often hard (or infeasible) to determine this task-specific metric. Nevertheless, in this section we are mainly interested in understanding the relationship between the notion of individual fairness and its connection to accuracy parity, where we use the Wasserstein distance as a bridge to connect these two. In particular, we say that a predictor $h$ to be individually fair if it treats similar individuals (measured by $d_T(\cdot, \cdot)$) similarly:

**Definition 3.2** (Individual Fairness, (Dwork et al., 2012)). For the task $T$ with task-specific metric $d_T(\cdot, \cdot)$, a regressor $h$ satisfies $\rho$-individual fairness if $\forall x, x' \in \mathcal{X}, |h(x) - h(x')| \leq \rho d_T(x, x')$.

Essentially, individual fairness puts a Lipschitz continuity constraint on the predictor with respect to the task-specific metric $d_T(\cdot, \cdot)$. Note that in the original definition (Dwork et al., 2012, Definition 2.1) the authors use a general metric

$d_T(\cdot, \cdot)$ as a similarity measure between individuals, and the choice of such similarity measure is at the center of related applications. In this section we use $\|\cdot\|$ in Definition 3.2 mainly for the purpose of illustration, but the following results can be straightforwardly extended for any metric $d_T(\cdot, \cdot)$. Another notion of group fairness that has gained increasing attention (Buolamwini & Gebru, 2018; Bagdasaryan et al., 2019; Chi et al., 2021) is *accuracy parity*:

**Definition 3.3** ($\epsilon$-Accuracy Parity). Given a joint distribution $\mu$ and $0 \le \epsilon \le 1$, a regressor $\widehat{Y} = h(X)$ satisfies $\epsilon$-*accuracy parity* if $|\varepsilon_{1,\mu_0}(\widehat{Y}) - \varepsilon_{1,\mu_1}(\widehat{Y})| \le \epsilon$.

Accuracy parity calls for approximately equalized performance of the predictor across different groups. The following proposition states the relationship between individual fairness, accuracy parity and the $W_1$ distance between the distributions $\mu_0$ and $\mu_1$ of different groups:

**Proposition 3.2.** If $h(\cdot)$ is $\rho$-individually fair, then $h$ satisfies $\sqrt{\rho^2 + 1} \cdot W_1(\mu_0, \mu_1)$-accuracy parity.

Together with Lemma A.1 in the appendix, Proposition 3.2 suggests that in order to achieve approximate accuracy parity, one can constrain the predictor to be Lipschitz continuous while at the same time try to decrease the $W_1$ distance between the distributions across groups, via learning representations. In the case where the groups are similar and the Wasserstein distance is small, individual fairness provides some guidance towards approximate accuracy parity. However, in cases where the groups are different (disjoint), the representation learning becomes more important.

### 3.4 Fair Representations with Wasserstein Distance

From the previous discussion, we know that the Wasserstein distance between the predicted distributions and the input distributions can be used to control both statistical parity and accuracy parity, respectively. *Is there a way to simultaneously achieve both goals?* In this section we shall provide an affirmative answer to this question via learning fair representations. The high-level idea is quite simple and intuitive: given input variable $X$, we seek to learn a representation $Z = g(X)$ such that $W_1(g_\sharp\mu_0, g_\sharp\mu_1)$ is small. If furthermore the predictor $h$ acting on the representation $Z$ is individually fair, we can then hope to have small statistical and accuracy disparity simultaneously.

Concretely, the following proposition says if the Wasserstein distance between feature distributions from two groups, $W_1(g_\sharp\mu_0, g_\sharp\mu_1)$, is small, then as long as the predictor is individually fair, it also satisfies approximate statistical parity:

**Proposition 3.3.** Let $Z = g(X)$ be the features from input $X$. If $W_1(g_\sharp\mu_0, g_\sharp\mu_1) \le \epsilon$ and $\widehat{Y} = h(Z)$ is $\rho$-Lipschitz, then $\widehat{Y} = (h \circ g)(X)$ verifies $2\sqrt{C\rho\epsilon}$-SP.

In practice since we only have finite samples from the corresponding distributions, we will replace all the distributions with their corresponding empirical versions. Furthermore, instead of using the joint error as our objective function, as we discussed in the previous section, we propose to use the balanced error rate instead:

$$\min_{g, \|h\|_L \le \rho} \max_{\|f\|_L \le 1} \quad \frac{1}{2}\left(\varepsilon_{2,\mu_0}(h \circ g) + \varepsilon_{2,\mu_1}(h \circ g)\right) + \tau \cdot \left|\mathbb{E}_{g_\sharp\mu_0}[f(Z)] - \mathbb{E}_{g_\sharp\mu_1}[f(Z)]\right|, \tag{10}$$

where $\tau > 0$ is a hyperparameter that trades off the $\ell_p$ error and the Wasserstein distance and $\rho$ is the Lipschitz constant of the predictor. The above problem could be optimized using the gradient descent-ascent algorithm (Edwards & Storkey, 2015; Zhang et al., 2018). To implement the Lipschitz constraints, we apply weight clipping to the parameters of both the adversary as well as the target predictor. More specifically, we use the projected gradient descent algorithm to ensure the $\ell_2$ norm of the discriminator and the target predictor to be bounded by the preset values.

**Comparisons with Related Fair Representation Learning Method**  One closely related approach that uses the Wasserstein distance as a penalty term in representation learning is from Chi et al. (2021), where the authors also formulated a minimax approach to encourage accuracy disparity between different groups. Compared with Eq. (10), from a model perspective, the main difference between our formulation and the one in Chi et al. (2021) is that the discriminator in Eq. (10) only takes the features $Z$ from different groups as input whereas the corresponding discriminator in Chi et al. (2021) takes both the features $Z$ and the label $Y$ as input. Because of this difference, the motivations of these two approaches are quite different. In our case, we use the Wasserstein distance to penalize the discrepancy between the marginal feature distributions in order to (approximately) achieve statistical parity, whereas

Table 1: Fair representations with Wasserstein regularization on the Law School dataset. We report the overall error, group-wise error, statistical disparity, and accuracy disparity.

| | $\tau$ | $\varepsilon_\mu$ | $\varepsilon_{\mu_0} + \varepsilon_{\mu_1}$ | $K(\widehat{Y}_0, \widehat{Y}_1)$ | $|\varepsilon_{\mu_0}(\widehat{Y}) - \varepsilon_{\mu_1}(\widehat{Y})|$ |
|---|---|---|---|---|---|
| MLP | N/A | $0.034_{\pm 0.005}$ | $0.069_{\pm 0.011}$ | $0.296_{\pm 0.044}$ | $0.011_{\pm 0.004}$ |
| W-MLP | 0.1 | $0.034_{\pm 0.004}$ | $0.067_{\pm 0.008}$ | $0.222_{\pm 0.037}$ | $0.011_{\pm 0.003}$ |
| W-MLP | 1.0 | $0.030_{\pm 0.000}$ | $0.059_{\pm 0.001}$ | $0.116_{\pm 0.032}$ | $0.011_{\pm 0.002}$ |
| W-MLP | 5.0 | $0.034_{\pm 0.002}$ | $0.067_{\pm 0.004}$ | $0.084_{\pm 0.073}$ | $0.008_{\pm 0.002}$ |
| W-MLP | 10.0 | $0.035_{\pm 0.001}$ | $0.069_{\pm 0.003}$ | $0.048_{\pm 0.059}$ | $0.006_{\pm 0.000}$ |

the Wasserstein metric used in Chi et al. (2021) is used to align the conditional feature distributions (conditioned on the label $Y$) between different groups, in order to minimize the accuracy disparity. To encourage accuracy parity, in Eq. (10), we also constrain the predictor to be Lipschitz continuous, following Proposition 3.2.

# 4 Experiments

Our theoretical results imply that even if there is no significant drop in terms of the overall population error when a model is built to satisfy the statistical parity, the minority group can still suffer greatly from the reduction in accuracy. On the other hand, by using the balanced error rate as the objective function, we can mitigate the disparate drops in terms of accuracy between these two groups. Furthermore, by minimizing the Wasserstein distance of the feature distributions across groups, we can hope to achieve both approximate statistical and accuracy parity. To verify these implications, we conduct experiments on a real-world benchmark dataset, the Law School dataset (Wightman, 1998), to present empirical results with various metrics. We refer readers to Appendix B for further details about the dataset, our pre-processing pipeline and the models used in the experiments.

**Experimental Setup** To demonstrate the effect of using Wasserstein distance to regularize the representations with adversarial training, we perform a controlled experiment by fixing the baseline model to be a three hidden-layer feed-forward network with ReLU activations, denoted as MLP. We use W-MLP to denote the model with Wasserstein constraint for representation learning. In the experiment, all the other factors are fixed to be the same across these two methods, including learning rate, optimization algorithm, training epoch, and also batch size. To see how the Wasserstein regularization affects the joint error, the conditional errors as well as the statistical parity and accuracy parity, we vary the coefficient $\tau$ for the adversarial loss between 0.1, 1.0, 5.0 and 10.0. For each experiment, we repeat each experiment for 5 times and report both the mean and the error bars.

**Results and Analysis** The experimental results are listed in Table 1. In the table we use $K(\widehat{Y}_0, \widehat{Y}_1)$ to denote the Kolmogorov-Smirnov distance of the predicted distribution across groups, which is also the value of approximate statistical parity. From the table, it is then clear that with increasing $\tau$, both the statistical disparity and the accuracy disparity are decreasing. Interestingly, the overall error $\varepsilon_\mu$ (sensitive to the distribution of $A$) and the sum of group errors $\varepsilon_{\mu_0} + \varepsilon_{\mu_1}$ (insensitive to the imbalance of $A$) only marginally increase. In fact, for $\tau = 1.0$, we actually observed better accuracy. We conjecture that the improved performance stems from the implicit regularization via weight clipping of the target predictor. With $\tau = 10.0$, the last row shows that this method could effectively reduce both the statistical and accuracy disparity to a value very close to 0, although at the cost of increasing errors. To conclude, all the empirical results are consistent with our findings.

# 5 Related Work

**Fair Regression** Two central notions of fairness have been extensively studied, i.e., individual fairness and group fairness. Dwork et al. (2012) defined individual fairness as a Lipschitz constraint of the underlying (randomized) algorithm. However, this definition requires a priori a distance metric to compute the similarity between pairs of individuals, which is often hard to construct or design in practice. Group fairness is a statistical definition, and it includes a family of definitions which essentially ask some statistical scores to be equalized between different subgroups. Typical examples include statistical parity, equalized odds (Hardt et al., 2016), and accuracy parity (Buolamwini &

Gebru, 2018). In this work we focus on an extension of statistical parity to regression problems, and study its theoretical tradeoff with accuracy, when the regressor cannot directly take the protected attribute as input during both training and inference stages. We also investigate the relationship between individual fairness and accuracy parity, and provide a bound through the Wasserstein distance. The line of work on fair regression through regularization techniques dates at least back to Calders et al. (2013), where the authors enforce a first-order moment requirement between the predicted distributions. More recent works include (Komiyama et al., 2018) that uses coefficient of determination as a notion of fairness when there are multiple sensitive attributes. When the sensitive attribute is continuous, generalized definition using the Rényi correlation coefficient exists (Mary et al., 2019). In a recent work, Agarwal et al. (2019) proposed a reduction approach from fair regression to a sequence of cost-sensitive minimization problems. Other approaches include a two-stage recalibration procedure (Chzhen et al., 2020b) and robust optimization techniques (Narasimhan et al., 2020). Our definition of statistical parity in the regression setting is stronger than the one of Calders et al. (2013), which proposed to use the difference of means as the metric. When the output dimension is one, our definition also coincides with the one proposed by Agarwal et al. (2019), which amounts to the Kolmogorov-Smirnov distance.

**Tradeoff between Fairness and Accuracy**     Although it has long been empirically observed that there is an inherent tradeoff between accuracy and statistical parity in both classification and regression problems (Calders et al., 2009; Zafar et al., 2015; Zliobaite, 2015; Berk et al., 2017; Corbett-Davies et al., 2017; Zhao et al., 2019a), precise characterizations on such tradeoffs are less explored. Berk et al. (2017) defined the price of fairness (PoF) under their convex framework for linear and logistic regression problems as the ratio between the loss of the optimal (approximately) fair regressor and the Bayes optimal loss. Under this PoF definition, the authors then explored the accuracy-fairness frontier by changing the approximate coefficient of the fairness constraint. Menon & Williamson (2018, Proposition 8) explored such tradeoff in terms of the fairness frontier function under the context of cost-sensitive binary classification. Zhao & Gordon (2022) proved a lower bound on the joint error that has to be incurred by any fair algorithm satisfying statistical parity. Our negative result is similar to that of Zhao & Gordon (2022) in nature, and could be understood as a generalization of their results from classification to regression. Chzhen et al. (2020a) and Le Gouic et al. (2020) concurrently with each other derived an analytic bound to characterize the price of statistical parity in regression when the learner can take the sensitive attribute explicitly as an input for $\ell_2$ loss. In this case, the lower bound is given by the optimal transport distance from two group distributions to a common one, characterized by the $W_2$ barycenter. Our results differ in that in our setting the learner cannot use the sensitive attribute directly as an input, and our results hold for the general $\ell_p$ loss. Note that this is significant, because it is not clear how to extend the results to space with $W_p$ as a metric, since the proof depends on the use of the Pythagoras' decomposition, which only holds under the $\ell_2$ distance.

On the upside, under certain data generative assumptions of the sampling bias, there is a line of recent works showing that fairness constraints could instead improve the accuracy of the predictor (Dutta et al., 2020; Blum & Stangl, 2020). In particular, Blum & Stangl (2020) prove that if the observable data are subject to labeling bias, then the Equality of Opportunity constraint could help recover the Bayes optimal classifier. Note that this does not contradict with our results, since in this work we do not make any assumptions on the underlying training distributions.

# 6    Conclusion

In this paper we show that when the target distribution differs across different demographic subgroups, any attribute-blind fair algorithm in the statistical parity sense has to achieve a large error on at least one of the groups. In particular, we give a characterization of such tradeoff using the Wasserstein distance. On the other hand, we also establish a connection between individual fairness and accuracy parity, where again, the accuracy disparity gap is characterized by the Wasserstein distance. Besides the theoretical contributions, our analysis using Wasserstein distance also suggests a practical algorithm for fair regression through learning representations for different demographic subgroups that are close in the sense of Wasserstein distance. Empirical results on a real-world dataset also confirm our findings.

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

# A Missing Proofs

In this section we provide all the missing proofs in the main text. For the ease of the readers, in what follows we shall first restate the theorems that appear in the main text and then provide the corresponding proofs.

## A.1 Proofs of Theorem 3.1 and Corollary 3.1

**Theorem 3.1.** Let $\widehat{Y} = h(X)$ be a predictor. If $\widehat{Y}$ satisfies statistical parity, then $\forall p \geq 1$,

$$\varepsilon_{p,\mu_0}(\widehat{Y}) + \varepsilon_{p,\mu_1}(\widehat{Y}) \geq W_p(Y_\sharp \mu_0, Y_\sharp \mu_1). \tag{5}$$

*Proof.* First, realize that $W_p(\cdot, \cdot)$ is a metric of probability distributions, the following chain of triangle inequalities holds:

$$W_p(Y_\sharp \mu_0, Y_\sharp \mu_1) \leq W_p(Y_\sharp \mu_0, h_\sharp \mu_0) + W_p(h_\sharp \mu_0, h_\sharp \mu_1) + W_p(h_\sharp \mu_1, Y_\sharp \mu_1).$$

Now due to the assumption that $\widehat{Y} = h(X)$ is independent of $A$, the second term, $W_p(h_\sharp \mu_0, h_\sharp \mu_1)$, is 0, leading to:

$$W_p(Y_\sharp \mu_0, Y_\sharp \mu_1) \leq W_p(Y_\sharp \mu_0, h_\sharp \mu_0) + W_p(h_\sharp \mu_1, Y_\sharp \mu_1). \tag{11}$$

Next, for $a \in \{0, 1\}$, by definition of the Wasserstein distance,

$$W_p(Y_\sharp \mu_a, h_\sharp \mu_a) = \left( \inf_\gamma \mathbb{E}_\gamma [|Y - \widehat{Y}|^p] \right)^{1/p} \leq \left( \mathbb{E}_{\mu_a} [|Y - \widehat{Y}|^p] \right)^{1/p} = \varepsilon_{p,\mu_a}(\widehat{Y}), \tag{12}$$

where we use the fact that the pushforward distribution of $\mu_a$ under $h$ is a particular coupling between $Y$ and $\widehat{Y}$ to establish the above inequality. Applying the inequality (12) for both $a = 0$ and $a = 1$ and combining it with inequality (11) completes the proof. ∎

**Corollary 3.1.** If $\widehat{Y}$ satisfies statistical parity, then $\varepsilon_{1,\mu_0}(\widehat{Y}) + \varepsilon_{1,\mu_1}(\widehat{Y}) \geq |\mathbb{E}_{\mu_0}[Y] - \mathbb{E}_{\mu_1}[Y]|$ and $\varepsilon_{2,\mu_0}^2(\widehat{Y}) + \varepsilon_{2,\mu_1}^2(\widehat{Y}) \geq \frac{1}{2}|\mathbb{E}_{\mu_0}[Y] - \mathbb{E}_{\mu_1}[Y]|^2$.

*Proof.* We first prove the first inequality in Corollary 3.1. Apply Theorem 3.1 by setting $p = 1$. Let $\text{Id} : \mathcal{Y} \to \mathcal{Y}$ be the identity map, i.e., $\text{Id}(y) = y$, $\forall y \in \mathbb{R}$. Clearly $\text{Id}(\cdot)$ is 1-Lipschitz. Using the sup characterization of the Wasserstein distance, we have:

$$\mathbb{E}_{\mu_0}[|\widehat{Y} - Y|] + \mathbb{E}_{\mu_1}[|\widehat{Y} - Y|] \geq W_1(Y_\sharp \mu_0, Y_\sharp \mu_1) \qquad \text{(Theorem 3.1)}$$

$$= \sup_{\|f\|_L \leq 1} \left| \int f \, d(Y_\sharp \mu_0) - \int f \, d(Y_\sharp \mu_1) \right|$$

$$\geq \left| \int \text{Id} \, d(Y_\sharp \mu_0) - \int \text{Id} \, d(Y_\sharp \mu_1) \right| \qquad \text{(Id is 1-Lipschitz)}$$

$$= \left| \int Y \, d\mu_0 - \int Y \, d\mu_1 \right| \qquad \text{(Change of Variable)}$$

$$= |\mathbb{E}_{\mu_0}[Y] - \mathbb{E}_{\mu_1}[Y]|, \tag{13}$$

where the second to last equation follows from the definition of pushforward distribution.

To prove the second inequality in Corollary 3.1, again set $p = 2$ in Theorem 3.1 and realize that for $a, b$, we have $a^2 + b^2 \geq (a + b)^2/2$ by the AM-GM inequality. Hence when $p = 2$, we have

$$\mathbb{E}_{\mu_0}[|\widehat{Y} - Y|^2] + \mathbb{E}_{\mu_1}[|\widehat{Y} - Y|^2] \geq \mathbb{E}_{\mu_0}^2[|\widehat{Y} - Y|] + \mathbb{E}_{\mu_1}^2[|\widehat{Y} - Y|] \quad \text{(Jensen's inequality)}$$

$$\geq 2\left(\frac{\mathbb{E}_{\mu_0}[|\widehat{Y} - Y|] + \mathbb{E}_{\mu_1}[|\widehat{Y} - Y|]}{2}\right)^2 \quad \text{(AM-GM inequality)}$$

$$= \frac{1}{2}\left(\mathbb{E}_{\mu_0}[|\widehat{Y} - Y|] + \mathbb{E}_{\mu_1}[|\widehat{Y} - Y|]\right)^2$$

$$\geq \frac{1}{2}|\mathbb{E}_{\mu_0}[Y] - \mathbb{E}_{\mu_1}[Y]|^2. \quad \text{(Eq. (13))}$$

Note that for the last equation we apply the lower bound $\mathbb{E}_{\mu_0}[|\widehat{Y} - Y|] + \mathbb{E}_{\mu_1}[|\widehat{Y} - Y|] \geq |\mathbb{E}_{\mu_0}[Y] - \mathbb{E}_{\mu_1}[Y]|$ we proved in the first inequality. ∎

## A.2 Proof of Theorem 3.2

Before we provide the proof of Theorem 3.2, we first recall some useful results about the Wasserstein distance (Weed et al., 2019; Lei et al., 2020).

**Proposition A.1** (Proposition 20, (Weed et al., 2019))**.** For all $n \geq 0$ and $p \geq 1$, let $\hat{\mu}$ be an empirical distribution induced from $\mu$ with sample size $n$. Then,

$$\Pr(W_p^p(\mu, \hat{\mu}) \geq \mathbb{E}W_p^p(\mu, \hat{\mu}_n) + t) \leq \exp(-2nt^2). \tag{14}$$

Proposition A.1 gives a concentration inequality of $W_p^p(\cdot, \cdot)$ around its mean. Note that the expectation in (14) is over the draw of the sample of size $n$. This inequality is particularly useful when $p = 1$ since it reduces to $W_1(\cdot, \cdot)$ and gives a convergence rate of $O(1/\sqrt{n})$.

The following theorem is a special case of (Lei et al., 2020, Theorem 3.1), which bounds the rate of $\mathbb{E}W_p(\mu, \hat{\mu}_n)$:

**Theorem A.1** (Theorem 3.1, (Lei et al., 2020))**.** Let $\hat{\mu}$ be an empirical distribution induced from $\mu$ with sample size $n$, $p \geq 1$ and recall that $\mathcal{Y} = [-1, 1]$. Then

$$\mathbb{E}W_p(Y_\sharp\mu, Y_\sharp\hat{\mu}) \leq c_p \cdot n^{-\frac{1}{2p}}, \tag{15}$$

where $c_p$ is a positive constant that only depends on $p$.

Again, the interesting case here is when $p = 1$, which gives the same rate of $O(1/\sqrt{n})$ that coincides with the one in Proposition A.1.

**Theorem 3.2.** Let $\widehat{Y} = h(X)$ be the predictor and $\hat{\mu}$ be an empirical distribution induced from a sample of size $n$ drawn from $\mu$. If $\widehat{Y}$ satisfies statistical parity, then there exists an absolute constant $c_1 > 0$ such that for $0 < \delta < 1$, with probability at least $1 - \delta$ over the draw of the sample,

$$\varepsilon_{2,\mu_0}(\widehat{Y}) + \varepsilon_{2,\mu_1}(\widehat{Y}) \geq \varepsilon_{1,\mu_0}(\widehat{Y}) + \varepsilon_{1,\mu_1}(\widehat{Y}) \geq W_1(Y_\sharp\hat{\mu}_0, Y_\sharp\hat{\mu}_1) - \left(2c_1 + \sqrt{2\log(2/\delta)}\right)\sqrt{\frac{1}{n}}. \tag{6}$$

*Proof.* We first prove the finite sample lower bound w.r.t. the $\ell_1$ error. Realize that $W_1(\cdot, \cdot)$ is a metric, the triangle inequality gives us

$$W_1(Y_\sharp\hat{\mu}_0, Y_\sharp\hat{\mu}_1) \leq W_1(Y_\sharp\hat{\mu}_0, Y_\sharp\mu_0) + W_1(Y_\sharp\mu_0, Y_\sharp\mu_1) + W_1(Y_\sharp\mu_1, Y_\sharp\hat{\mu}_1).$$

Combined with Theorem 3.1, the above inequality leads to

$$\varepsilon_{1,\mu_0}(\widehat{Y}) + \varepsilon_{1,\mu_1}(\widehat{Y}) \geq W_1(Y_\sharp\hat{\mu}_0, Y_\sharp\hat{\mu}_1) - \left(W_1(Y_\sharp\hat{\mu}_0, Y_\sharp\mu_0) + W_1(Y_\sharp\mu_1, Y_\sharp\hat{\mu}_1)\right).$$

Hence it suffices if we could provide high probability bound to further lower bound $W_1(Y_\sharp\mu_i, Y_\sharp\hat{\mu}_i)$, for $i \in \{0, 1\}$. To this end, we first apply Proposition A.1 with $p = 1$: let $\exp(-2nt^2) = \delta/2$ and solve for $t$, we have $t = \sqrt{\log(2/\delta)/2n}$, which means that with probability at least $1 - \delta/2$,

$$
\begin{aligned}
W_1(Y_\sharp\hat{\mu}_i, Y_\sharp\mu_i) &\leq \mathbb{E}W_1(Y_\sharp\hat{\mu}_i, Y_\sharp\mu_i) + \sqrt{\frac{\log(2/\delta)}{2n}} \\
&\leq c_p\sqrt{\frac{1}{n}} + \sqrt{\frac{\log(2/\delta)}{2n}}.
\end{aligned}
\quad \text{(Theorem A.1)}
$$

Now apply the above inequality twice, one for $i \in \{0, 1\}$. With a union bound, we have shown that w.p. $\geq 1 - \delta$,

$$
\varepsilon_{1,\mu_0}(\widehat{Y}) + \varepsilon_{1,\mu_1}(\widehat{Y}) \geq W_1(Y_\sharp\hat{\mu}_0, Y_\sharp\hat{\mu}_1) - \left(2c_1 + \sqrt{2\log(2/\delta)}\right)\sqrt{\frac{1}{n}}.
$$

To prove the second lower bound w.r.t. the $\ell_2$ error, simply realize that $\varepsilon_{2,\mu_i}(\widehat{Y}) \geq \varepsilon_{1,\mu_i}(\widehat{Y})$ for $i \in \{0, 1\}$, which completes the proof. ∎

### A.3 Proof of Corollary 3.2

**Corollary 3.2.** Let $\widehat{Y} = h(X)$ be a predictor. If $\widehat{Y}$ satisfies statistical parity, then $\forall p \geq 1$, the joint error has the following lower bound:

$$
\varepsilon_{p,\mu}(\widehat{Y}) \geq H_{\text{0-1}}(A) \cdot W_p(Y_\sharp\mu_0, Y_\sharp\mu_1). \tag{7}
$$

*Proof.* To simplify the notation used in the proof, define $\varepsilon := \varepsilon_{p,\mu}(\widehat{Y})$, $\varepsilon_0 := \varepsilon_{p,\mu_0}(\widehat{Y})$ and $\varepsilon_1 := \varepsilon_{p,\mu_1}(\widehat{Y})$. Let $p_a := \Pr_\mu(A = a)$. By Theorem 3.1, we know that $\varepsilon_0 + \varepsilon_1 \geq W_p(Y_\sharp\mu_0, Y_\sharp\mu_1)$. By definition of the joint error:

$$
\varepsilon = p_0\varepsilon_0 + p_1\varepsilon_1 \geq \min\{p_0, p_1\}(\varepsilon_0 + \varepsilon_1) \geq H_{\text{0-1}}(A) \cdot W_p(Y_\sharp\mu_0, Y_\sharp\mu_1). \qquad \blacksquare
$$

### A.4 Proof of Proposition 3.1

As a comparison to the Kolmogorov-Smirnov distance, the $W_1$ distance between distributions over $\mathbb{R}$ could be equivalently represented as:

**Proposition A.2** (Gibbs & Su (2002)). For two distributions $\mu, \mu'$ over $\mathbb{R}$, $W_1(\mu, \mu') = \int_{\mathbb{R}} |F_\mu(z) - F_{\mu'}(z)| \, dz$.

Proposition A.2 was stated as a fact without proof in (Gibbs & Su, 2002), but it is not hard to see that it could be proved using the equivalent characterization of $W_1$ in (4) by changing the integral variable. Furthermore, in regression if both $\mu$ and $\mu'$ are continuous distributions, then the following well-known result (Chatterjee, 2007, Lemma 2) serves as a bridge to connect the Wasserstein distance $W_1(\cdot, \cdot)$ and the Kolmogorov-Smirnov distance $K(\cdot, \cdot)$:

**Lemma A.1.** If there exists a constant $C$ such that the density of $\mu'$ (w.r.t. the Lebesgue measure $\lambda$) is universally bounded above, i.e., $\|d\mu'/d\lambda\|_\infty \leq C$, then $K(\mu, \mu') \leq 2\sqrt{C \cdot W_1(\mu, \mu')}$.

Using Kolmogorov-Smirnov distance, the constraint in the optimization problem (2) could be equivalently expressed as $K(h_\sharp\mu_0, h_\sharp\mu_1) \leq \epsilon$. Now with Lemma A.1, we are ready to prove Proposition 3.1:

**Proposition 3.1.** Let $\widehat{Y} = h(X)$ be a predictor. Under Assumption 2.1, for $p \geq 1$, if there exists $\epsilon > 0$ such that $W_p(h_\sharp\mu_0, h_\sharp\mu_1) \leq \epsilon$, then

$$
r_{p,\mu_0}(\widehat{Y}) + r_{p,\mu_1}(\widehat{Y}) \geq \underbrace{W_p(f^*_{0\sharp}\mu_0, f^*_{1\sharp}\mu_1)}_{\text{distance between optimal unfair predictors across groups}} -2(\varepsilon^*_{p,\mu_0} + \varepsilon^*_{p,\mu_1}) - \epsilon, \tag{9}
$$

and $\widehat{Y}$ satisfies $2\sqrt{C\epsilon}$-SP.

*Proof.* First, for $a \in \{0,1\}$, by definition of the Wasserstein distance, for any predictor $\widehat{Y} = h(X)$:

$$W_p(Y_\sharp \mu_a, h_\sharp \mu_a) = \left( \inf_\gamma \mathbb{E}_\gamma [|Y - \widehat{Y}|^p] \right)^{1/p} \le \left( \mathbb{E}_{\mu_a} [|Y - \widehat{Y}|^p] \right)^{1/p} = \varepsilon_{p,\mu_a}(\widehat{Y}), \tag{16}$$

Applying the above inequality to both $h$ and $f_a^*$, we have:

$$W_p(h_\sharp \mu_a, Y_\sharp \mu_a) + W_p(Y_\sharp \mu_a, f_a^*{}_\sharp \mu_a) \le \varepsilon_{p,\mu_a}(\widehat{Y}) + \varepsilon_{p,\mu_a}(f_a^*(X)) = \varepsilon_{p,\mu_a}(\widehat{Y}) + \varepsilon_{p,\mu_a}^*, \quad \forall a \in \{0,1\}. \tag{17}$$

On the other hand, by the triangle inequality,

$$W_p(h_\sharp \mu_0, h_\sharp \mu_1) + \sum_{a \in \{0,1\}} W_p(h_\sharp \mu_a, Y_\sharp \mu_a) + W_p(Y_\sharp \mu_a, f_a^*{}_\sharp \mu_a) \ge W_p(f_0^*{}_\sharp \mu_0, f_1^*{}_\sharp \mu_1).$$

Now by the assumption $W_p(h_\sharp \mu_0, h_\sharp \mu_1) \le \epsilon$ and Eq. (17), we have:

$$\epsilon + \sum_{a \in \{0,1\}} \varepsilon_{p,\mu_a}(\widehat{Y}) + \varepsilon_{p,\mu_a}^* \ge W_p(f_0^*{}_\sharp \mu_0, f_1^*{}_\sharp \mu_1).$$

By the definition of the excess risk, rearranging and subtracting $2\sum_{a \in \{0,1\}} \varepsilon_{p,\mu_a}^*$ from both sides of the inequality then completes the proof of the first part.

To show that $\widehat{Y}$ is $2\sqrt{C\epsilon}$-SP, first note that $\widehat{Y} = h(X)$ is $t$-SP iff $K(h_\sharp \mu_0, h_\sharp \mu_1) \le t$. Now apply Lemma A.1, under the assumption that $W_p(h_\sharp \mu_0, h_\sharp \mu_1) \le \epsilon$:

$$K(h_\sharp \mu_0, h_\sharp \mu_1) \le 2\sqrt{CW_1(h_\sharp \mu_0, h_\sharp \mu_1)} \qquad \text{(Lemma A.1)}$$

$$\le 2\sqrt{CW_p(h_\sharp \mu_0, h_\sharp \mu_1)} \qquad \text{(Monotonicity of the } W_p(\cdot, \cdot))$$

$$\le 2\sqrt{C\epsilon},$$

completing the proof. ∎

## A.5 Proof of Proposition 3.2

**Proposition 3.2.** If $h(\cdot)$ is $\rho$-individually fair, then $h$ satisfies $\sqrt{\rho^2 + 1} \cdot W_1(\mu_0, \mu_1)$-accuracy parity.

*Proof.* Define $g(X, Y) := |\widehat{Y} - Y| = |h(X) - Y|$. We first show that if $h(X)$ is $\rho$-Lipschitz, then $g(X, Y)$ is $\sqrt{\rho^2 + 1}$-Lipschitz: for $\forall x, y, x, x'$:

$$\begin{aligned}|g(x, y) - g(x', y')| &= ||h(x) - y| - |h(x') - y'|| \\ &\le |h(x) - h(x') - y + y'| \qquad &\text{(Triangle inequality)} \\ &\le |h(x) - h(x')| + |y - y'| \\ &\le \rho \|x - x'\| + |y - y'| \qquad &(h \text{ is } \rho\text{-Lipschitz}) \\ &\le \sqrt{\rho^2 + 1} \cdot \sqrt{\|x - x'\|^2 + |y - y'|^2} \qquad &\text{(Cauchy-Schwarz)} \\ &= \sqrt{\rho^2 + 1} \cdot \|(x, y) - (x', y')\|.\end{aligned}$$

Let $\rho' := \sqrt{\rho^2 + 1}$. Now consider the error difference:

$$\begin{aligned}|\varepsilon_{1,\mu_0}(\widehat{Y}) - \varepsilon_{1,\mu_1}(\widehat{Y})| &= |\mathbb{E}_{\mu_0}[|h(X) - Y|] - \mathbb{E}_{\mu_1}[|h(X) - Y|]| \\ &= |\mathbb{E}_{\mu_0}[g(X, Y)] - \mathbb{E}_{\mu_1}[g(X, Y)]| \\ &\le \sup_{\|g'\|_L \le \rho'} |\mathbb{E}_{\mu_0}[g'(X, Y)] - \mathbb{E}_{\mu_1}[g'(X, Y)]| \\ &= \rho' \sup_{\|g'\|_L \le 1} |\mathbb{E}_{\mu_0}[g'(X, Y)] - \mathbb{E}_{\mu_1}[g'(X, Y)]| \\ &= \rho' \cdot W_1(\mu_0, \mu_1), \qquad &\text{(Kantorovich duality)}\end{aligned}$$

which completes the proof. ∎

### A.6 Proof of Proposition 3.3

**Proposition 3.3.** Let $Z = g(X)$ be the features from input $X$. If $W_1(g_\sharp\mu_0, g_\sharp\mu_1) \leq \epsilon$ and $\widehat{Y} = h(Z)$ is $\rho$-Lipschitz, then $\widehat{Y} = (h \circ g)(X)$ verifies $2\sqrt{C\rho\epsilon}$-SP.

*Proof.* We first show that $W_1((h \circ g)_\sharp\mu_0, (h \circ g)_\sharp\mu_1)$ is small if $h$ is $\rho$-Lipschitz. To simplify the notation, we define $\mu_0' := g_\sharp\mu_0$ and $\mu_1' := g_\sharp\mu_1$. Consider the dual representation of the Wasserstein distance:

$$
\begin{aligned}
W_1(h_\sharp\mu_0', h_\sharp\mu_1') &= \sup_{\|f'\|_L \leq 1} \left| \int f'\, d(h_\sharp\mu_0') - \int f'\, d(h_\sharp\mu_1') \right| && \text{(Kantorovich duality)} \\
&= \sup_{\|f'\|_L \leq 1} \left| \int f' \circ h\, d\mu_0' - \int f' \circ h\, d\mu_1' \right| && \text{(Change of Variable formula)} \\
&\leq \sup_{\|f\|_L \leq \rho} \left| \int f\, d\mu_0' - \int f\, d\mu_1' \right| && \text{($h$ is $\rho$-Lipschitz)} \\
&= \rho \cdot W_1(\mu_0', \mu_1') \\
&= \rho \cdot W_1(g_\sharp\mu_0, g_\sharp\mu_1) \\
&\leq \rho\epsilon,
\end{aligned}
$$

where the first inequality is due to the fact that for $\|f'\|_L \leq 1$, $\|f' \circ h\|_L \leq \|f'\|_L \cdot \|h\|_L = \rho$. Applying Lemma A.1 to $W_1(h_\sharp\mu_0', h_\sharp\mu_1')$ then completes the proof. ∎

## B Further Details about the Experiments

### B.1 Dataset

The Law School dataset contains 1,823 records for law students who took the bar passage study for Law School Admission.[2] The features in the dataset include variables such as undergraduate GPA, LSAT score, full-time status, family income, gender, etc. In the experiment, we use gender as the protected attribute and undergraduate GPA as the target variable. We use 80 percent of the data as our training set and the rest 20 percent as the test set. The data distribution for different subgroups in the Law School dataset could be found in Figure 2. In the Law School dataset, $\Pr(A = 1) = 0.452$, which is a quite balanced dataset. All the experiments are performed on a Titan 1080 GPU.

### B.2 Network Architectures

We fix the baseline model to be a three hidden-layer feed-forward network with ReLU activations. The number of units in each hidden layer are 50 and 20, respectively. The output layer corresponds to a linear regression model. This baseline is denoted as MLP. For learning with Wasserstein regularization, the adversarial discriminator network takes the feature from the last hidden layer as input, and connects it to a hidden-layer with 10 units, followed by an auditor whose goal is to output a score function in distinguishing the features from the two different groups. This model is denoted as W-MLP. Compared with MLP, the only difference of W-MLP in terms of objective function is that besides the $\ell_2$ loss for target prediction, the W-MLP also contains a loss from the auditor to distinguish the sensitive attribute $A$.

### B.3 Hyperparameters used in Experiments

In this section we report the detailed hyperparameters used in our experiments to obtain the results in Table 1. Throughout the experiments, we fix the learning rate to be 1.0 and use the same networks as well as random seeds. One important aspect in the implementation of the Wasserstein adversary is the choice of the clipping parameter for the weights in the adversary network. The values used in our experiments are shown below in Table 2.

---

[2]We use the edited public version of the dataset which can be downloaded here: `https://github.com/algowatchpenn/GerryFair/blob/master/dataset/lawschool.csv`

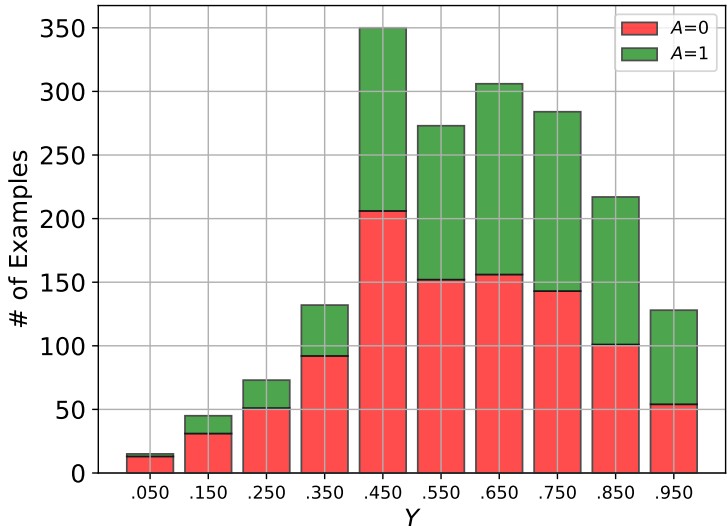

Figure 2: The data distributions of different groups in the Law School dataset.

Table 2: Clipping parameters used in training the Wasserstein adversary.

|        | $\tau$ | Clipping Value |
|--------|--------|----------------|
| W-MLP  | 0.1    | 0.1            |
| W-MLP  | 1.0    | 1.0            |
| W-MLP  | 5.0    | 5.0            |
| W-MLP  | 10.0   | 10.0           |

