# OpenReview forum: "Costs and Benefits of Fair Regression"
_TMLR — Rejected by TMLR_

### Review · Reviewer_bCn9 · 2022-05-10

**Summary Of Contributions:**

The paper studies the trade-off between accuracy and fairness, as measured by statistical parity, in the context of regression with binary protected attributes. In contrast to previous work, the authors consider a setup where the predictors do not take the protected attribute as an input.

The authors prove a lower bound on the sum of the expected losses of any fair classifier on the two protected subgroups. This lower bound depends on a Wasserstein distance between the data distributions conditioned on the value of the protected attribute. The result also implies a lower bound on the global accuracy of fair classifiers, that additionally depends on the 0-1 entropy of the protected attribute. The results are extended to the case when a classifier is only approximately fair.

Additionally, the authors show that any individually fair classifier satisfies a certain amount of accuracy parity fairness, with the amount again depending on a Wasserstein distance notion between the protected subpopulations. The result extends to the case of representation learning and the authors use this to design an algorithm for training with accuracy parity constraints. The algorithm is tested on the Law School dataset.

**Broader Impact Concerns:**

Nothing comes to mind currently.

**Requested Changes:**

Major points:

As pointed it out above, I find the comparison to Chzhen et al. (2020) and Le Gouic et al. (2020), in its current version, very insufficient and potentially technically incorrect and/or misleading. Therefore, I believe that my concerns above should be addressed in order for the paper to be ready for acceptance.

Minor points:

Below are some typos and things that can be clarified.

Page 2

- It might be nice to add a citation as soon as accuracy parity is introduced (right after individual fairness)
- Defining $A$, $A$ is in $\{ 0,1 \}$, not in $\{ 0,1 \}^1$
- Defining the cumulative distribution function of $\mu$, formally the probability is taken in the product space of $X \times A \times Y$

Page 3

- What is the difference between $\mu_{Y}$ and $Y_{\#}\mu$?
- Assumption 2.1: why use $\mu '$, not $\mu$

Page 7

- First line in Section 3.2: "In the last section" --> "In the previous section"


**Strengths And Weaknesses:**

Strengths:

- The paper is mathematically sound, excluding minor typos/notational confusions, see "Requested changes"
- The paper addresses a broad audience, since a lot of current effort in ML research is dedicated to ML fairness and therefore understanding the limits of fairness-aware learning is certainly a very relevant problem.
- The presented links between individual fairness, accuracy parity and the Wasserstein distance are, to my awareness, also novel and certainly interesting.

Weaknesses:

My most significant concern is the comparison with the papers of Chzhen et al. (2020) and Le Gouic et al. (2020). Both of these papers clearly address a very closely related problem. The only difference, as I understand, is that in their case the classifier also takes the protected attributes as an input.

The authors compare to these results in the "informal" proposition 3.1. I went through the statement and its proof and I am concerned about the following issues:

1. The authors state that the cost of fairness is higher in previous works, as compared to this one. The only reason for this informal statement, as presented in the proof, is that the lower bound in the previous papers was larger than the one in this paper. However, it could well be that the bound in this paper (in Corollary 3.2) is loose and therefore it only appears as if not taking the protected attribute as an input helps for a better fairness-accuracy trade-off.

2. Given that the lower bound from Chzhen et al. (2020) is larger than the one in this paper, I actually believe that the bound of Chzhen et al. (2020) implies Corollary 3.2. In particular, take any fair $h(X)$ and define $g(X, A) = h(X)$. This is a perfectly valid function on $X \times A$ and it is also fair, because $g(X, A) = h(X)$ and $h(X)$ is independent of $A$, therefore $g(X, A)$ is also independent of $A$. Therefore, I believe that the result of Chzhen et al. (2020) automatically implies a better lower bound.

Please do correct me if I'm missing something here.

3. The presented proof really does not seem to match the intuition presented in the main body immediately after Proposition 3.1. The intuitive statement claims that the trade-off between fairness and accuracy is worse when the protected attribute is taken as an input, because in such a case the optimal possible accuracy is higher and so there is more to be lost. However, the proof simply shows that the lower bound of Chzhen et al. (2020) is larger and I do not see how this is related to the optimal accuracy being higher.

Additionally, I’m not sure whether these other papers really count as “concurrent work” (as stated in the related work section), since they are both almost two years old.

---

### Review · Reviewer_xobk · 2022-05-15

**Summary Of Contributions:**

The paper considers $\ell_p$ regression $X\in \mathbb{R}^d \to Y\in[-1,1]$, while constraining for statistical parity, i.e., predictions $\hat Y$ are required to be (approximately) independent of a binary protected attribute $A$. The main result(s) are lower bounds on the sum of $A$-conditional (excess) errors, based on the approximate independence of the (noisy) target $Y$ itself from $A$. This approximate independence is characterized via Wasserstein distances. These distances are then used in a couple of tangential contexts: to characterize the relationship between individual fairness and accuracy parity and to suggest fair representation learning to simultaneously achieve accuracy parity and statistical parity.

**Broader Impact Concerns:**

No Broader Impact Statement present. No concerns.

**Requested Changes:**

- (Sec. 2) In the notations, restrict the use of $z$ for features (similarly in Eq. 2, use $y$). Also, $\mu$ is introduced as the overall underlying measure, but then it is also used as a placeholder for any measure; use a different letter for that. Use $a$ consistently for the protected attribute, sometimes $i$ is also used. Assumption 2.1 repeats what is already in the text, remove one or the other.
- (Sec 3.1) I suggest clearly discussing the role of noiselessness and attribute-blindness (or lack thereof). The claims of tightness must be tempered, to reflect what is given, or strengthened, via further examples. The comparison between attribute-blindness and attribute-awareness (Prop. 3.1) should be entirely removed, as it is factually incorrect.
- (Sec 3.1) Make it clear that the $\ell_2$ case follows immediately from prior results (Theorem A.2)
- (Sec 3.2) Do not claim Prop. 3.2 is the reason why you could use Wasserstein distance on inputs for distribution parity, but that rather it’s Lemma A.1.
- (Sec 3.4) Do not claim that the lower bounds have any bearing on the use of the Wasserstein distance for fair representation learning. Make it clear that the latter does not guarantee achieving the lower bound.
- (Sec 4) Give ample detail of the actual losses used, especially on the adversarial network side, and clearly relate to Eq. (10).


**Strengths And Weaknesses:**

## Strengths
- The problem of understanding the tradeoffs of introducing fairness constraints in statistics and machine learning is important and timely. This paper tackles this in somewhat more generality than previous work.
- The paper is technically sound, with the analysis presented clearly and methodically.

## Weaknesses
- (Sec. 2) Example 1 doesn’t quite serve its purpose, because its conclusion holds for any $h$ that depends only on $X$. In other words, fairness does not factor in. This violates precisely what the authors try to contrast in the introduction when they say (Sec. 1):
> Hence it highlights the inherent tradeoff due to the coupling between statistical parity and accuracy in general, not due to the noninformativeness of the input.
- (Sec 3.1) A first claimed differentiating aspects of the paper is that it applies to learners that do not have access to the protected attribute. However, the proof of Thm. 3.1 works almost identically when the protected attribute is accessible, by using $h_0$ and $h_1$ in each case, and the conclusion remains unchanged. Thus, Thm. 3.1’s lower bound is not specifically tailored to attribute-blind learners and does not separate the attribute-blind and attribute-aware cases. (Even the caption of Fig. 1 makes it clear that $h_0$ and $h_1$ can be different, which shows the authors know this and might have made a strategic choice of marketing this as a differentiating aspect.)
- (Sec 3.1) A second claim of the paper is that the presented lower bound is tight. When a lower bound has a varying parameter, then to properly claim it is tight, examples must be constructed for every choice of the parameter. In this case, $p$ is a parameter, but the only tight example that is given is for $p=2$. So, either do not make the tightness claim, or span the range of all $p$. (This is also important because, see below, the $p=2$ case follows directly from prior results.)
- (Sec 3.1) A third claim of the paper is that, by imposing fairness, attribute-aware learners lose more accuracy than attribute-blind ones (Prop. 3.1). However, this line of argument is flawed. The explanation in the main text seems to suggest comparing loss differentials, which may make sense (but not because “the two search spaces become the same”, they do not). The analytic approach in the appendix, however, deviates from this in two flawed ways: (1) compares losses directly (not differentials) and (2) uses lower bounds for this comparison. The problem is that the lower bound of Cor. 3.2 can be very loose, so saying we’re not suffering as much based on that is incorrect. Also, two lower bounds can be tight on two very different examples, so it’s not an apples-to-apples comparison. Effectively App. A.4 only shows that the bound of Cor. 3.2 is weaker than previously known Thm. A.2.
- (Sec 3.1) Speaking of Thm. A.2, it is important to note that by using the fact that $v^*$ is the geodesic midpoint (through a Pythagorean inequality), we get the $\ell_2$ version of Thm. 3.1 directly. (This same property can also gain a factor of $2$ in the “proof” of Prop 3.1).  This applies even for $h$ that doesn’t depend on $A$, because $\min_h \geq min_g$ in the theorem. It is important to clarify that the novelty of the paper is only for other values of $p$.
- (Sec 3.2 to Sec 3.3) The paper makes a couple of clunky transitions between the main results and the ancillary results. The first is at the end of Sec. 3.2, where Prop 3.2 is said to imply that Wasserstein distance is a good proxy to approximate statistical parity. The implication comes rather from the previously known result of Lemma A.1, that upper bounds KS distance with the (square root of the) $\ell_1$-Wasserstein distance. (Prop 3.2 is just an algorithm-independent lower bound and does not demonstrate the proxy-fitness in specific approaches, e.g., solving an empirical version of Eq. (2) perhaps with KS swapped for Wasserstein, similarly to the suggestion of Eq. (10).)
- (Sec.3.3 to Sec 3.4) The second clunky transition occurs at the beginning of Sec. 3.4 where the appearance of Wasserstein distance in multiple places (“key role”) is touted as a unifying concept. But these are very different appearances: _(i)_ $W(Y_\sharp\mu_0,Y_\sharp\mu_1)$ the distance between targets, _(ii)_ $W(f^*_{0\sharp}\mu_0,f^*_{1_\sharp}\mu_1)$ the distance between optimal regressors, _(iii)_ $W(\mu_0,\mu_1)$, the distance between populations; all attribute-conditional. These are very different quantities, and in Sec. 3.4 a fourth one is introduced, $W(g_\sharp\mu_0,g_\sharp\mu_1)$, the distance between features. Most importantly, it is implicitly implied that the lower bounds have some bearing on this choice for fair representation learning. But this won’t alter targets or optimal regressors, the only way it acts is by directly forcing the predictors to not vary across attributes, which is the definition of statistical parity. In other words, none of the lower bound theory is relevant to this, and we could have arrived immediately to this point through a relaxation of the Kolmogorov-Smirnov distance (see also previous point).
- (Sec 3.4) Prop 3.4 says nothing about what kind of accuracy one could expect under these conditions. One would hope that the inspiration of the lower bound would compel the design an algorithm that helps reach it, as suggested in the abstract. But Prop 3.4 and Eq. (10) fall short of this goal.
- (Sec 4 and B.3) The adversarial network and particularly the corresponding loss is an important aspect of the experiments that is not well described. It seems that the authors use a variation of the GAN discriminator loss, which isn’t quite an implementation of Eq. (10). It is important to make this explicit and to compare in case of any deviation from (10). I believe, for example, that the GAN loss does not reduce to expectations of founded Lipschitz functions, because of the logarithm.

### Minor comments:
- A more informative title could be chosen, e.g., Accuracy Cost of Fairness in $\ell_p$ Regression via Wasserstein Distances
- (Sec. 1) Even though the Wasserstein distances are useful in this context, I am not sure we can conclude that they are “central”. Independence can be characterized in many ways, and one example at which the lower bounds are tight does not mean another characterization won’t be as good.
- (Ex. 3.2), the distribution of $X_\sharp\mu_a$ not depending on $a$ already implies that $A\perp X$.
- (Sec. 3.1) The presentation does not make it clear where we use the noiselessness (existence of perfect predictors of $Y$). In fact, I think we don’t.
- (Sec 3.1) In the remark prior to Cor. 3.2, the Wasserstein distance is not _estimated_, but rather _computed_ efficiently. It’s already an empirical quantity, so it is just a matter of computing it with an efficient algorithm.
- (Sec 3.1) The proof of Thm. 3.2 can be easily extended to give a $1/\sqrt{n}$ bound even with $W_p$ (all the $p$’s in the powers simplify). Why not include that version instead?
- (Sec 3.1) In the proofs, it is worth having a consistent shortcut for $Pr(A=0)$ (sometimes $\alpha$  is used, other times $p_0$ is used).
- (Sec 3.2) Why not call the optimal errors and regressors Bayes optimal? Also,the $f_a^*$ may be achievable, but may not be unique. Either assume unique or replace $=$ with $\in$ in front of the $\arg\min$’s, and change *an* ~~the~~ optimal.
- (Sec 3.2) In Prop 3.2, it is not clear why we restrict the Bayes optimal regressors to depend only on $X$ (attribute-blind)
- (Sec 3.3) A proof of Prop 3.3 with the primal form of Wasserstein distance is much shorter: couple $X,Y\sim \mu_0$ and $X’,Y’\sim \mu_1$, note that $|h(X)-Y|-|h(X’)-Y’|$ $< |h(X)-h(X’)| + |Y-Y’|$ $< \rho d(X,X’) + |Y’-Y|$ $< \sqrt{\rho^2+1} (\sum_i |X_i-X’_i|+|Y-Y’|)$, then use the primal form to replace with the Wasserstein distance.
- (Sec 4) Why not use LSAT scores for predictions? The task of predicting undergrad GPA (backward in time) does not sound too natural.
- (A.3) The proof of Cor. 3.2 is overly complicated. You get the answer in a single line: $\epsilon = \alpha \epsilon_0 +(1-\alpha) \epsilon_1 \geq (\alpha \wedge 1-\alpha) (\epsilon_0+\epsilon_1)$.
- (B.3) Why are clipping values identical to $\tau$?

### Typos:
- there exist~~s~~ (unfair) algorithms
- benefits of ~~for~~ fair regression, our analysis ~~using~~ also naturally
- $\epsilon$-SP, Definition ~~3.2~~ *2.2*
- Proof of (Cor. 3.2), the AM-GM inequality gives $a^2+b^2 \geq \mathbf{\frac{1}{2}} (a+b)^2$.

---

### Review · Reviewer_Ukxq · 2022-06-07

**Summary Of Contributions:**

Summary:
   The paper derives algorithm independent lower bounds on the balanced error rate of fair classifiers where the fairness criterion being studied is statistical parity, i.e. the sensitive attribute and the classifier output must be independent. The lower bound is in terms of how correlated target Y is with the group indicator measured in terms of distributions of target conditioned on the group indicator.

Assuming that the sensitive attribute is not directly used in the classification and the actual target is a deterministic function of $X$, suppose the fair classifier achieves exact statistical parity (measured in the Kolmogorov Smirnov metric), then author show that the balanced error rate in the p-norm (sum of errors of the classifier on each of the sensitive groups) is lower bounded by Wasserstein p-distance between the target variable across groups. These bounds are shown to be tight as well.

Authors derive corollaries with a finite sample version where distances are measured with respect to the empirical measures under both sub groups. Further, they also consider the unbalanced error rate with different weighings.

Authors then derive lower bounds when statistical parity is achieved approximately and when target $Y$ is a noisy version of feature $X$ where they lower bound the sum of excess risk from the best classifier on each group by Wasserstein distance between marginal distribution of the best classifiers output across the groups.

Authors also show how representation learning - where Wasserstein distance between a representation 'g' is approximately matched across the groups, any Lipschitz classifier built on top of it  satisfies approximate accuracy parity (measured in the $\ell_p$ norm) and statistical parity. Authors then propose to learn a classifier $h \circ g$ where $g$ is a representation that matches across sensitive groups and $h$ optimizes the balanced error rate across the groups. They also experimentally demonstrate that representation learning simultaneously achieves the accuracy parity and statistical parity.






**Broader Impact Concerns:**

One simple concern is that this paper's claims is limited to accuracy parity and statistical parity. In many fairness applications, that may not be the best criterion to optimize for - conditional statistical parity , Path specific fairness etc.. would be desirable for which the recommendations in this paper may not apply. This is not a big concern but something that could be noted.

One issue is that, these optimization procedure might work in domain. Consider the covariate shift scenario where P(Y|X) remains the same while P(X,G) (where G is the sensitive attribute) shifts in the test distribution. Would the fairness guarantees hold domains out of distribution ?
In reality, one of the issues could be that any fair classifier even with guarantees gets hobbled by distributional shift. For example P (X|G) could shift in a way to make the representations learnt not match anymore. So it would make sense to use these prescriptions carefully when test domain shifts.

**Requested Changes:**

Changes:
What would have been nice if authors compared their representation learning approach to a simpler baseline where balanced error rate is minimized subject to two different regularizers - one optimising accuracy parity and the other statistical parity. Will the doubly constrained formulation yield a better tradeoff point. (repeating the point from the weakness section) than representation learning ?



**Strengths And Weaknesses:**

Strengths:
    Algorithm independent lower bounds on fair classifiers that exactly or approximately satisfies statistical parity. In some cases, the lower bounds are shown to be tight.

 Further, showing that representation learning to match the groups and then learning a classifier on top can simultaneously achieve accuracy parity and statistical parity.

I find these two main messages are new, technically elegant and very interesting for the fairness community.

I went through the proofs and they appear correct to me.

Weaknesses:
        a) What would have been nice if authors compared their representation learning approach to a simpler baseline where balanced error rate is minimized subject to two different regularizers - one optimising accuracy parity and the other statistical parity. Will the doubly constrained formulation yield a better tradeoff point than something attained through representation learning ?
 What is not clear is the "necessity" of representation learning in Section 3.3.  Can authors comment on it ? or perhaps add the above baseline (if its not computationally too difficult).
     b) This is not a weakness but a question, the main results for wasserstein (like Theorem 3.1) seem to follow from triangle inequality for Wasserstein distance. For classification problems, is there something that can be said about errors in total variation distance or KL loss that is tight ?

I lean positively since the theoretical results seem technically novel and elegant (to my knowledge) shedding light on issues around statistical parity and accuracy parity.

---

### Review · Reviewer_TuEw · 2022-06-10

**Summary Of Contributions:**

This paper provides theoretical results concerning fair regression in the case where the regressor cannot take the sensitive attribute as input. The first set of results are lower bounds on the sum of group-wise $\ell_p$ errors when subject to statistical parity. Variations include "noiseless" vs. "noisy" distributions, approximate statistical parity, and a finite-sample bound. An additional result (Proposition 3.3) connects individual fairness, accuracy parity, and the 1-Wasserstein distance between group-conditional distributions. Based on the appearance of Wasserstein distance in the theoretical results, a representation learning approach is proposed to reduce statistical disparity and accuracy disparity simultaneously by minimizing the Wasserstein distance between group-conditional representations. This approach is briefly evaluated on one dataset.

**Broader Impact Concerns:**

None.

**Requested Changes:**

Please address the major weaknesses discussed above. Concretely:
- Discuss the relationship with Chi et al. (ICML 2021) at an overall level
- Relate Theorem 3.1, Proposition 3.2 in the current paper to Theorem 3.1 of Chi et al.
- Discuss the similarities and differences between problem (10) in the current paper and the Wasserstein variant in Section 3.2 of Chi et al. Perform an empirical comparison if possible.
- Rewrite Proposition 3.1 as a precise mathematical result comparing Corollary 3.2 to Theorem 2.3 of Chzhen et al. (2020).
- Ensure that textual descriptions of Proposition 3.1 are faithful.
- Qualify phrases like "any fair regressor", etc.
- Make the review of the fair regression literature in Section 5 more complete.

Minor Comments:
- Fix any errors, in particular the missing Lipschitz constraint in (10).
- Addressing the other comments would solidify proofs and improve clarity.

**Strengths And Weaknesses:**

**Strengths:**

+ Addresses fundamental, algorithm-independent trade-offs in fair regression subject to statistical parity
+ Proposition 3.3 is particularly noteworthy as it relates individual fairness and the group-based notion of accuracy parity
+ The results are straightforward to state and interpret, and the proofs are not hard to follow for the most part

**Major Weaknesses:**

1. My main concern is that the paper has similarities with Chi et al. (ICML 2021), which is cited, but these similarities are not acknowledged, let alone discussed in satisfactory detail. There is considerable overlap in setup (fair regression without sensitive attribute as input, accuracy parity) and techniques (e.g. use of Wasserstein distance and triangle inequalities involving Wasserstein). Thus there needs to be a detailed discussion/comparison of the relationship and benefits of the results in the present paper compared to those of Chi et al. Concretely:

    1. Theorem 3.1 in Chi et al. is also a lower bound on the sum of group-wise errors in terms of Wasserstein distance. How does it relate to Theorem 3.1, Proposition 3.2, etc., in the current paper? One clear difference is that Chi et al. use 1-Wasserstein versus 2-Wasserstein, but beyond that it is not clear.

    2. Problem formulation (10) in the current paper is quite similar to the Wasserstein variant in Section 3.2 of Chi et al., in that both propose a Wasserstein penalty on the group-conditional induced distributions. I believe the differences are that a) $h$ is constrained to be Lipschitz, although this constraint is missing from (10), and b) Chi et al.'s discriminator also takes $Y$ as input. At a minimum, these similarities and differences should be described. Even better would be a discussion of the implications of the differences, and better still an empirical comparison with the Wasserstein variant of Chi et al. in the experiment of Section 4. The latter would show whether (10) is indeed providing a new, superior capability to reduce statistical disparity and accuracy disparity simultaneously.

2. Comparison of the lower bounds of Corollary 3.2 and Chzhen et al. (2020), Theorem 2.3: I happen to be the fourth reviewer to submit a review, even though my review is by no means late. So I can see (and unfortunately cannot unsee) the comments made by two of the reviewers about this comparison, and I do not disagree with their excellent points. I will confine my comments to presentation:
    1. By labelling Proposition 3.1 as "informal" and giving little mathematical detail in the main paper, I almost concluded that the comparison remained at the level of hand-waving, when in fact an analytical comparison of lower bounds has been done (although how this result should be sold is under debate). So my recommendation is to state the precise mathematical result in the main paper, and then ensure that the textual descriptions of this result are faithful to it and do not oversell.
    2. There are multiple instances of phrases like "any fair regressor," "any fairness algorithm," "quantitative characterization on the exact tradeoff between fairness and accuracy is still missing," which should really be qualified in light of Chzhen et al.'s and Le Gouic et al.'s prior work. Qualifiers like "attribute-blind" and "for all $\ell_p$ norms" come to mind.

3. Since the paper is on fair regression, the literature review in Section 5 should be more complete. Some works (Berk et al. (2017); Chi et al. (2021); Chzhen et al. (2020)) are cited but not discussed in context in Section 5. Missing works include:
    - Komiyama et al., ICML 2018
    - Mary et al., ICML 2019
    - Narasimhan et al., AAAI 2020
    - Chzhen et al., NeurIPS 2020, "Fair regression via plug-in estimator and recalibration with statistical guarantees"

**Minor Comments:**

- The introduction could state up front that the paper addresses multiple fairness notions, to prepare the reader.
- Introduction, last paragraph, "benefits of for fair regression, our analysis using also": Missing or extra words here.
- Assumption 2.1: What is the meaning of $\mu'$ as distinct from $\mu$? Also this assumption makes the sentence above it redundant.
- Example 3.2, the first displayed equation with two inequalities: I do not see the justification for the first inequality.
- Proposition 3.2: Should state that Assumption 2.1 is used here, in part to remind readers about $C$.
- Eq. (10): I believe $h$ is missing a Lipschitz constraint.
- Section 3.4, last sentence: This is not very clear about how weight clipping is applied to enforce Lipschitzness of $f$ and presumably $h$ ("target predictor"?). Appendix B.2 also does not describe this. I reviewed a previous version of this paper in which more detail was given, which I feel should be added back.
- Appendix A.4, "price when regressor cannot access $A$": I believe $W_p$ should be $W_2$.
- Lemma A.1: This "well-known result" needs a reference.

---

### Decision · Action_Editors · 2022-08-08

**Recommendation:** Reject

**Comment:**

As all reviewers agreed, this paper addressed an important problem in fair regression, and provided some insights in the trade-off between fairness vs. accuracy, which could be a potential important contribution to the community.

During the discussion period, the authors successfully address most of the concerns and questions raised by the reviewers, and make the contribution of the paper more clear, especially introducing direct comparison with existing works (Chzhen et al. 2020 and Chi et al. 2021).

However, there are still several issues raised by reviewers, which are needed to be addressed:

* The author revised the objective function, and discussed the weighted average $\ell_p$ loss, which is quite different from the existing works. It will be great if the authors can emphasize the differences and provide more justification about the choice of such an objective function.

* Since the objective function has been modified in the updated version, some of the reviewers are concerning about the consistency and coherence of the submission. It will be great if the authors can recheck the whole paper and ensure the consistency.

* In technical detail part, the results should be discussed more comprehensively. For example, as raised by Reviewer xobk,
    *  explain how the p=2 case, even with their sum-of-errors objective, is a simple corollary from past results;
    *  explain whether these results specialize to attribute-blindness/aware or not;
    *  provide more motivations for the Wassertein based representation from the first part analysis of the paper;
    *  the tightness of the lower bound besides the extreme case.

Therefore, most of the reviewers would like to have another round of full review. However, since in TMLR decision system, there is no "major revision", I would like to encourage the authors to consider to resubmit the manuscript to TMLR after these revisions.